# ADVERSARIAL POLICY TRANSFER IN MIXED COOPERATIVE-COMPETITIVE GAMES

## ABSTRACT

Equilibrium learning in mixed cooperative–competitive games remains a central challenge, as empirical algorithms often become trapped in suboptimal or locally stable equilibria. Adversarial policies expose vulnerabilities in such equilibria learned by victim agents through task-irrelevant actions, a problem well studied in two-agent zero-sum games and only recently extended to multi-agent reinforcement learning (MARL). Existing approaches often overfit to specific scenarios and lack generalization, considering task-specific vulnerabilities only and requiring millions of interactions to adapt to new settings, which is impractical in real world. By contrast, transferable MARL methods can generalize across tasks but focus on overpowering opponents rather than strategically exploiting their weaknesses. Here we propose a transferable adversarial policy framework for mixed cooperative–competitive games that enables zero-shot attacks in previously unseen scenarios, revealing the existence of shared vulnerabilities in learned MARL policies and enabling efficient and accurate robustness assessment without training a separate attack for each policy. Our approach has two key components. First, *adversarial tactic acquisition* iteratively extracts attack strategies that reliably deceive victim agents, using large language models (LLMs) during training and Bayesian inference to weight tactics at test time. Second, *adversarial scene decomposition* partitions attack scenarios into smaller, transferable subgames that consistently elicit adversarial behaviour, based on the interactions between attacker and victim teams. We provide a convergence proof alongside our approach. Empirically, we demonstrate adversarial policy transfer in *StarCraft II* and *MAgent* across 20 tasks with up to 64 victim agents, varying in number, type and policy. Training against our attack addresses common vulnerabilities in victim policies and enhances robustness to subsequent re-attacks.

## 1 INTRODUCTION

Mixed cooperative-competitive games involve two teams of agents that cooperate internally while competing against each other (Lowe et al., 2017; Xu et al., 2023; Carminati et al., 2022; Zhang et al., 2022). Such games naturally arise in domains like real-time strategy (e.g., StarCraft II) (Samvelyan et al., 2019), swarm robotics (Hüttenrauch et al., 2019; Batra et al., 2021), and team sports simulations (Kurach et al., 2020). Computing equilibria in these settings is APX-hard, due to the combined challenges of inter-team competition, intra-team coordination, and partial observability (Celli & Gatti, 2018). Consequently, practical MARL methods often simplify the environment by training against rule-based opponents (Samvelyan et al., 2019; Yu et al., 2022), or approximate equilibria via self-play and related game-theoretic techniques (Yang et al., 2018; Carminati et al., 2022).

Since exact equilibria are rarely achieved, the learned victim policies often get trapped in suboptimal equilibria, allowing opponents to exploit these weaknesses and gain an undue advantage. This phenomenon is well-documented in two-agent zero-sum games under the notion of *adversarial policies* (Gleave et al., 2020; Wu et al., 2021; Guo et al., 2021), where an adversary wins not by changing the environment but by executing task-irrelevant actions that deceive its opponent, misleading it to act awkwardly and fail to complete the task. Recent work shows that such adversarial policies also arise in mixed cooperative-competitive games, extending beyond the two-agent setting (Ma et al., 2024). However, existing approaches typically require millions of interactions with victim agents during training, which is impractical for real-world deployment. Moreover, these policies often overfit to

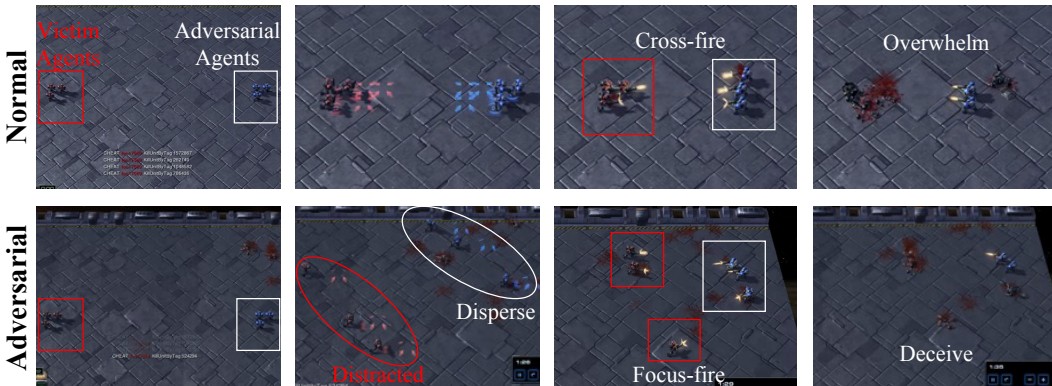

Figure 1: Comparison between transferable MARL (top) and transferable adversarial policies (bottom). While traditional MARL generally overwhelms opponents through direct confrontation, our method exploits vulnerabilities in learned policies via strategic, deceptive behavior.

specific victims and fail to exploit structural vulnerabilities that generalize across environments or policy types. To address this limitation, we propose training *transferable adversarial policies* that consistently deceive opponents across tasks with varying agent numbers, victim types, and policy classes, without additional fine-tuning. While related to transferable MARL (Hu et al., 2021; Qin et al., 2022; Tian et al., 2023), our approach is distinct: as illustrated in Fig. 1, prior work focuses on overpowering opponents, whereas we strategically exploit shared vulnerabilities in learned MARL policies. This leads to victims being scattered and unable to eliminate enemies quickly (Fig. 1), or even remaining stagnated and not moving or attacking at all (Fig. 4), highlighting the 'blind spots' in the policies.

In this paper, we show that transferable adversarial policies exist in MARL, exposing shared vulnerabilities across diverse learned policies. Our method further provides a practical tool for evaluating the robustness of mixed cooperative–competitive systems, enabling zero-shot deployment: once trained, the adversary can be applied directly to unseen scenarios without additional fine-tuning. We formalize it as a Bayesian Adversarial Zero-Sum Partially Observable Stochastic Game (BAZS-POSG), where uncertainty over the environment and victim agents is represented by latent embeddings encoding the attack tactics. Assuming fixed victim policies (Gleave et al., 2020), this reduces to a Bayesian Decentralized Partially Observable Markov Decision Process (Bayesian Dec-POMDP), in which adversarial agents cooperate to exploit the weaknesses of an unknown but fixed victim team.

Since the victim policy is unknown, it is challenging for transferable adversarial policies to exploit shared weaknesses across different victims. We address this through two components: (1) *adversarial tactic acquisition*, which learns generalizable tactics that deceive victims during training and provide high-level strategic guidance; and (2) *adversarial scene decomposition*, which partitions each scenario into smaller transferable subgames that consistently elicit adversarial behavior, offering low-level tactical guidance. For adversarial tactic acquisition, we leverage a large language model (LLM) to extract tactics from successful attack trajectories. These tactics serve as ground-truth labels for the latent embeddings associated with each environment–victim pair and are iteratively refined using rewards and trajectories from a transferable policy trained on previously proposed tactics. During each iteration, adversarial agents perform Bayesian inference (Harsanyi, 2004; Chen et al., 2021) to infer the LLM-provided tactics and condition their policies accordingly, enabling systematic reuse of effective strategies. For adversarial scene decomposition, we divide the game into subgames based on estimated interaction strength, capturing both intra-team cooperation and inter-team competition. These subgames localize adversarial behavior during transfer, particularly when structural patterns recur across tasks, thereby improving both generality and training efficiency. Alongside, we also provide a proof of the convergence of our approach. Empirically, we demonstrate effective adversarial policy transfer in StarCraft II and MAgent across 20 tasks, involving up to 64 victim agents with varying numbers, types, and policies. Training against our attack addresses common vulnerabilities in victim policies and enhances robustness to subsequent re-attacks.

**Contributions.** Our contributions are two-fold. First, we demonstrate that adversarial policy transfer in MARL is feasible, revealing shared vulnerabilities across different algorithms and scenarios.

Second, to enable this adversarial transfer, the adversary infers attack tactics via Bayesian inference, with optimal attack tactics iteratively refined by LLM. Next, the adversary decomposes new scenarios into smaller, previously seen subgames, enabling consistent adversarial behavior across tasks.

**Related work. Adversarial policies** have recently gained attention in MARL as a practical form of black-box attack that does not require access to victim parameters. In two-agent zero-sum games, Gleave *et al.* (Gleave et al., 2020) introduced adversarial policies that exploit neural policies through seemingly irrelevant yet deceptive actions. Follow-up work extended this idea to model-based planning (Wu et al., 2021), general-sum settings (Guo et al., 2021), and human-aligned behavior constraints (Bai et al., 2025). In cooperative MARL, adversarial behavior is often framed as one agent undermining its teammates through worst-case actions (Li et al., 2019; Lin et al., 2020), leading the team to suboptimal outcomes (Li et al., 2024; 2023a; Nisioti et al., 2021; Li et al., 2023b). The most relevant work to ours is SUB-PLAY (Ma et al., 2024), where the adversary exploits team-level weaknesses in mixed cooperative–competitive games by controlling an opposing team. However, existing adversarial policy methods require millions of direct interactions with victim agents, making them impractical in real-world settings. In contrast, we investigate whether adversarial policies can be made *transferable* in MARL, and how they generalize across environments and victim types with minimal or no direct interaction.

**Transfer learning in MARL** aims to reuse knowledge across tasks and can be broadly divided into two categories: network design and task embedding (Tian et al., 2023). Network design methods construct architectures that support cross-task generalization (Agarwal et al., 2020; Hu et al., 2021; Zhou et al., 2021; Zhang et al., 2023; Tian et al., 2023), including graph-based (Agarwal et al., 2020) and Transformer-based (Hu et al., 2021) population-invariant models, as well as hierarchical decision structures that capture patterns or skills (Zhou et al., 2021; Tian et al., 2023). Task embedding methods (Boutsioukis et al., 2011; Didi & Nitschke, 2016; Liu et al., 2019; Qin et al., 2022; Schäfer et al., 2022) instead learn latent representations of tasks to capture similarity. For instance, MATTER (Qin et al., 2022) assigns each task a basis-vector embedding and employs an explainer network for unseen tasks, while MATE (Schäfer et al., 2022) jointly learns the embedding space and explainer parameters. Although both approaches improve transferability, they typically target rule-based opponents. By contrast, our method transfers the ability to exploit vulnerabilities in learned policies.

## 2 PROBLEM FORMULATION

### 2.1 BAYESIAN ADVERSARIAL ZERO-SUM PARTIALLY OBSERVABLE STOCHASTIC GAME

We formulate our problem as a Bayesian Adversarial Zero-Sum Partially Observable Stochastic Game (BAZS-POSG). BAZS-POSG extends adversarial policy in multi-agent setting (Littman, 1994; Ma et al., 2024) by adding adversarial tactics to be used by adversaries, which serves as a latent embedding of the game. These embeddings act as unique identifiers that encode uncertainty over both the environment and the victim policies.

$$\mathcal{G} = \langle \mathcal{N}, \{\mathcal{T}_a, \mathcal{T}_v\}, \mathcal{S}, \mathcal{O}, O, \mathcal{A}, \Theta, \mathcal{P}, \mathcal{R}, \gamma \rangle. \tag{1}$$

Here $\mathcal{N} = \{1, ..., N\}$ is the set of $N$ agents in the game, $\{\mathcal{T}_a, \mathcal{T}_v\}$ partition $N$ agents into two teams, with $\{\mathcal{T}_a, \mathcal{T}_v\} \subseteq \mathcal{N}$, $\mathcal{T}_a \cup \mathcal{T}_v = \mathcal{N}$, $\mathcal{T}_a \cap \mathcal{T}_v = 0$. $i \in \mathcal{T}_a$ denotes the agent is an adversary, $i \in \mathcal{T}_v$ denotes the agent is a victim. $\mathcal{S}$ is the global state space. $\mathcal{O} = \times_{i \in \mathcal{N}} \mathcal{O}^i$ is the observation space of the agents, with $O$ the observation emission function. $\mathcal{A} = \times_{i \in \mathcal{N}} \mathcal{A}^i$ is the joint action space. $\Theta$ is the latent embedding that encodes difference in training and testing environments, including agent type, environment dynamics, *etc.* $\mathcal{P} : \mathcal{S} \times \mathcal{A} \times \Theta \to \Delta(\mathcal{S})$ is the state transition probability. For $i \in \mathcal{T}_a$, $\mathcal{R} : \mathcal{S} \times \mathcal{A} \times \Theta \to \mathbb{R}$ is the shared reward function for the adversaries. For $i \in \mathcal{T}_v$, the shared function for victims is $-\mathcal{R}$, corresponding to the zero-sum setting. $\gamma \in [0, 1)$ is the discount factor.

During training the transferable attacker, the embedding of game is sampled from $\theta \in \Theta_{train} \subseteq \Theta$, with $\Theta_{train}$ representing the set of embeddings in training environments. At each timestep $t$, victims and adversaries observe $o_t^i$ and adds it to their own history $h_t^i = [o_0^i, a_0^i, ..., o_t^i]$. Then, they make decisions using the policy of victims $\pi_v(a_t^i | h_t^i, \theta)$ and adversaries $\pi_a(a_t^i | h_t^i, \theta)$, forming a joint action $\mathbf{a}_t = \times_{i \in \mathcal{N}} \{a_t^i\}$. Here, we assume the policy of victims and adversaries condition on the current embedding $\theta$. The environment proceeds to next state following the transition probability $P(s_{t+1} | s_t, \mathbf{a}_t, \theta)$, and yields reward $r_t = R(s_t, \mathbf{a}_t, \theta)$ for adversaries and $-r_t$ for victims. The objective for adversary is to learn a transferable adversarial policy $\pi_a$ that maximize $r_t$ at test time,

with type $\theta \in \Theta_{test} \subseteq \Theta$, while $\Theta_{test}$ represents the set of embeddings in testing environments:

$$\max_{\pi_a} J(\pi_a) = \mathbb{E}_{\theta \in \Theta_{test}} \left[ \mathbb{E} \left[ \sum_{t=0}^{\infty} \gamma^t r_t | s_0 \sim \rho_0(\theta), \mathbf{a}_t \sim \prod_{i \in \mathcal{T}_a} \pi_a(\cdot | h_t^i, \theta) \prod_{j \in \mathcal{T}_v} \pi_v(\cdot | h_t^j, \theta) \right] \right]. \quad (2)$$

## 2.2 Simplification: Bayesian Dec-POMDP

Motivated by (Gleave et al., 2020; Ma et al., 2024), the victim policy is fixed after the embedding $\theta$ is assigned. Thus, the fixed victim policy can be merged in transition dynamics, resulting the zero-sum game between adversaries and victims to a cooperative game of adversaries itself. However, the challenge remains to identify and maximize the objective under unknown test-time environment. We formalize this as a *Bayesian Decentralized Partially Observable Markov Decision Process* (Bayesian Dec-POMDP), defined as a tuple:

$$\mathcal{G}_a = \langle \mathcal{T}_a, \mathcal{S}, \times_{i \in \mathcal{T}_a} \mathcal{O}^i, O, \times_{i \in \mathcal{T}_a} \mathcal{A}^i, \Theta, \mathcal{P}_a, \mathcal{R}, \gamma \rangle. \quad (3)$$

where $\mathcal{T}_a, \mathcal{S}, \mathcal{O}, O, \mathcal{A}, \Theta, \mathcal{R}, \gamma$ retain the same meanings as in Equation 2. $\mathcal{P}_a$ is the transition function of attackers, defined as $\sum_{\{a_t^i \in \mathcal{A}^i\}_{i \in \mathcal{T}_v}} P(s_{t+1}|s_t, \mathbf{a}_t) \prod_{i \in \mathcal{T}_v} \pi_v(a_t^i | h_t^i, \theta)$, since fixed victim policy can be treated as a part of environment transitions. In addition, we use the term "environment" to refer to the joint space of both the environment and the victim throughout the remainder of the paper.

## 2.3 Threat Model

Based on the challenges above, we further clarify the threat model, including the assumptions and capabilities of attackers and victims.

**Assumption 2.1 (Victim's assumption.)** Victims follow a fixed, well-trained learned policy $\pi_v$ parameterized by a neural network that remains unchanged during the attack.

The victim policy $\pi_v$ can be trained under two paradigms. The first trains victims against a fixed, rule-based opponent (*e.g.*, built-in rule-based AI in StarCraft II (Samvelyan et al., 2019)), a mainstream approach in cooperative MARL (Yu et al., 2022; Rashid et al., 2018). The second paradigm trains victims against another MARL opponent using game-theoretic approaches such as self-play (Yang et al., 2018; Xu et al., 2023). While these methods offer theoretical guarantees under idealized conditions, recent work in two-player zero-sum games shows that adversarial policies can still exploit them, winning easily by executing task-irrelevant yet deceptive actions (Gleave et al., 2020).

**Assumption 2.2 (Attacker's assumption.)** Attackers can train on a set of environments defined by latent embeddings $\theta \sim \Theta_{train}$, and have assess to global information, following the centralized training and decentralized execution (CTDE) paradigm (Rashid et al., 2018). During testing, attackers are evaluated on a set of environments that are not observed in training $\Theta_{test} \nsubseteq \Theta_{train}$.

To make transfer attack possible, we assume there exists shared and transferable information between training and testing. However, we define $\theta$ as a summary of all unknown hyperparameters during transfer, which remains unknown during both training and testing. Unlike standard assumptions in adversarial policy (Gleave et al., 2020; Ma et al., 2024), we assume no interaction with victims for training in previously unseen scenarios. While similar to the assumptions of transferable MARL, our attack additionally requires fooling victim agents during transfer, instead of overpowering them.

## 3 Adversarial Policy Transfer

In this section, we present our method for adversarial policy transfer in mixed cooperative-competitive games. In Section 3.1, the attackers perform adversarial tactic acquisition by using an LLM to extract high-level adversarial tactics in successful attack trajectories during training, and iteratively re-extract and refine them through subsequent attacks. At test time, attackers infer the optimal tactics via Bayesian inference to trigger similar vulnerabilities in victim behavior. In Section. 3.2, to enable attack behavior generalizing across diverse tasks, we introduce adversarial scene decomposition, which partitions attack scenarios into smaller, transferable subgames, based on inter- and intra-team

Figure 2: Attackers perform adversarial tactic acquisition by using LLMs to iteratively extract tactics and learn to approach them via Bayesian inference, enabling consistent tactic inference at test time. Then they apply adversarial scene decomposition to partition scenarios into transferable subgames based on inter- and intra-team interaction strength, enabling consistent behavior across tasks.

interaction strength between attackers and victims. This enables local elicitation of adversarial behavior during transfer, when similar attack structures are presented, providing low-level guidance for executing adversarial tactics. The training pipeline is illustrated in Fig. 2 with pseudo code in Appendix A.

## 3.1 ADVERSARIAL TACTIC ACQUISITION

Our formulation in Section 2.2 models adversarial transfer in mixed cooperative–competitive games as a Bayesian Dec-POMDP, where the uncertainty during transfer is represented as latent embeddings. Unlike prior approaches that attempt to model the environment and agents in full detail, we show that a compact representation, captured by what we term *adversarial tactics*, is sufficient. Empirically, adversaries deceive victims by collectively adopting such tactics during attacks. Examples include *retreat*, where adversaries withdraw to lure victims into pursuit, causing their policies to become dispersed and uncoordinated; and *cycling*, where adversaries evade firepower through continuous circling. These tactics reliably succeed against known victims, supporting their use as a sufficient representation for modeling unknown environments and victims during attacks.

We model tactics as a probability distribution $\Delta(\Lambda)$, where each element $\lambda \in \Lambda = \{1, ..., |\Lambda|\}$ denotes an individual tactic, such as *retreat* or *cycling*. The embeddings represent the proportions of tactics within a scenario. Because manually enumerating all possible tactics is infeasible, we adopt an LLM-driven approach (Fig. 3). For each training task, we train specialized adversarial policies against victim agents and translate the resulting trajectories into structured textual descriptions. An LLM (Chao et al., 2023; Wang et al., 2023) then summarizes the tactics from these adversarial trajectories and outputs their proportions across scenarios. We treat these outputs as labels, serving as ground truth embeddings within each iteration, for training adversarial policies under Bayesian inference. Tactics are re-summarized and refined iteratively. At each iteration, adversarial policies are trained from scratch on the latest LLM-provided labels, rather than fine-tuned, since the labels themselves change across iterations. The new trajectories and rewards are again translated into structured descriptions and supplied to the same LLM dialogue, which re-summarize the tactics and updates the tactic labels. From each iteration we select the policy with the best adversarial performance on the

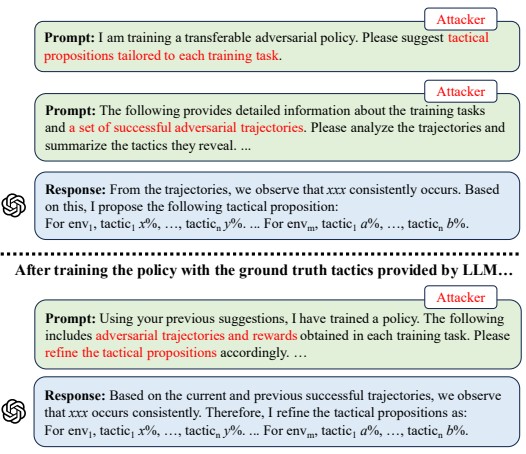

Figure 3: Illustration of adversarial tactic acquisition. We use an LLM to extract the high-level adversarial tactics in attack trajectories iteratively.

training tasks and deploy it directly to unseen target tasks without additional fine-tuning. Details of the LLM prompts and example outputs are provided in Appendix B.

Given adversarial tactics as sufficient information for each attack scenario, during training of the transferable attack, each agent learns a belief $b_t^i = [b_{t,1}^i, \ldots, b_{t,|\Lambda|}^i] \in \Delta(\Lambda)$ over the current tactics, equivalently the inference of the embedding of the current environment, and conditions its policy on it accordingly, where $b_{t,\lambda}^i$ donates the proportion assigned to tactic $\lambda$. To learn such beliefs, we treat the adversarial tactics provided by the LLM as ground-truth labels during training. Each agent infers the current tactic from the observed environment dynamics $P_\phi(o_{t+1}^i, r_t | h_t^i, a_t^i, \lambda)$ via Bayesian inference with a uniform prior $\mathcal{U}(\Lambda)$, where $\lambda \in \Lambda = \{1, \ldots, |\Lambda|\}$ represents a certain tactic:

$$b_{t+1,\lambda}^i = \frac{P_\phi(o_{t+1}^i, r_t | h_t^i, a_t^i, \lambda) b_{t,\lambda}^i}{\sum_{\lambda' \in \Lambda} P_\phi(o_{t+1}^i, r_t | h_t^i, a_t^i, \lambda') b_{t,\lambda'}^i}, b_0^i = \mathcal{U}(\Lambda). \tag{4}$$

We learn a model to represent current environment dynamics from collected trajectories, with adversarial tactics $\lambda_{LLM} = [\lambda_{LLM,1}, \ldots, \lambda_{LLM,|\Lambda|}] \in \Delta(\Lambda)$ given by LLM as ground-truth label:

$$\mathbb{E}_{(o_{t+1}^i, r_t, h_t^i, a_t^i) \sim D} - \log(\sum_{\lambda \in \Lambda} \lambda_{LLM,\lambda} P_\phi(o_{t+1}^i, r_t | h_t^i, a_t^i, \lambda)) \tag{5}$$

This design follows the CTDE paradigm. The dynamics model is trained centrally and later used during testing to infer optimal beliefs over adversarial tactics. To support transfer across varying populations and agent types, we construct the dynamics model using UPDeT (Hu et al., 2021) backbone and their observation decomposition pipeline. Specifically, we adopt a Transformer (Vaswani et al., 2017), treating $h_t^i, a_t^i$ as source tokens, $o_{t+1}^i, r_t$ as output tokens, with the output dimension corresponding to $|\Lambda|$. Notably, we do not incorporate any information from testing scenarios during the training phase, nor do we ask the LLMs to generate adversarial tactics for the testing scenarios. Besides, although LLMs may be trained on external sources that provide common-sense knowledge, we do not update the parameters of the LLMs. Once the model is trained, we deploy it zero-shot to the unseen testing scenarios. At these scenarios, each agent infers the proper tactics via Bayesian inference with the uniform prior, while the action-level policies remain fixed.

## 3.2 Adversarial Scene Decomposition

Adversarial tactic acquisition provides high-level strategic guidance to adversarial agents. However, agents may perceive the scenario differently across timesteps and contexts. To address this, we propose adversarial scene decomposition, which partitions the current scenario into smaller, transferable subgames (Zhou et al., 2021; Iqbal et al., 2021), patterns with only part of the agents compared to the original scenario, allowing local elicitation of adversarial behavior in test scenarios when similar subgames are encountered. Specifically, each adversarial agent can reduce its observation horizon, transforming its original observation into a familiar, localized view. This serves as a low-level guide, enabling agents to consistently deceive victim policies across tasks. Unlike prior work (Zhou et al., 2021; Iqbal et al., 2021), we integrate subgame partitioning directly into the decision process, improving interpretability.

To enable adversarial scene decomposition, we define the action space as $\mathcal{A}^i = \hat{\mathcal{A}}^i \times \mathcal{M}$, where $\hat{\mathcal{A}}^i$ represents the original game action space and $\mathcal{M} = \{0,1\}^{\mathcal{N}}$ indicates whether the current attacker chooses to perceive each other agent. Then, the actual policy used by attacker is: $\pi_a(a_t^i, m_t^i | h_t^i, b_t^i) = \pi_{\psi_1}(a_t^i | h_t^i, m_t^i, b_t^i) \prod_{j \in \mathcal{N}} \pi_{\psi_2}(m_t^{i,j} | h_t^i, b_t^i)$, where $m_t^i = \times_{j \in \mathcal{N}} m_t^{i,j}$ indicates whether each agent $j$ should be included in the subgame view of agent $i$, $\pi_{\psi_1}$ selects action with reduced view range, while $\pi_{\psi_2}$ determines which agent to block, parameterized by $\psi_1$ and $\psi_2$, respectively. This can be seen as partitioning current scenario into a set of subgames. Notably, the output mask includes both attackers and victims, modeling both inter- and intra-team interaction.

We use a Transformer-based, population-invariant backbone (Hu et al., 2021) for $\pi_{\psi_1}$ and $\pi_{\psi_2}$, where the belief $b_t^i$ is treated as an additional input token. To model the interaction with other agents, we decompose the observation of each agent as $o_t^i = [o_t^{i,1}, \ldots, o_t^{i,\mathcal{N}}]$. Thus, the representation of all other agents to agent $i$ is encoded as $[e_t^{i,1}, \ldots, e_t^{i,\mathcal{N}}] = f_{\psi_2}(h_t^i, b_t^i)$ using a self-attention mechanism (Vaswani et al., 2017), with $h_t^i$ denotes the history of agent $i$. Interaction strength between agents is

computed via cosine similarity and normalized to the range $[0, 1]$:

$$\pi_{\psi_2}(m^{i,j} = 1 | h_t^i, b_t^i) = \left( \frac{e_t^{i,i} \cdot e_t^{i,j}}{|e_t^{i,i}| \cdot |e_t^{i,j}|} + 1 \right) / 2 \tag{6}$$

After the masks $m_t^i$ are determined, if $m_t^{i,j} = 0$, the corresponding observation $o_t^{i,j}$ is excluded from the self-attention mechanism in $\pi_{\psi_1}$, thereby forming subgames by limiting interactions between agents. To ensure the local-global consistency of our adversarial scene decomposition, we add all intermediate decisions, including $b_t^i$ and $m_t^i$, to the input of the Q function. The convergence of Q function thus ensures global optimality. Assume the posterior belief $b_t^i \doteq p(\theta | h_t^i)$ over type $\theta$ is updated by Bayes' rule in Eqn. 4, the Q function can be defined as:

$$Q^i(s, \mathbf{a}, \mathbf{m}, b) = \mathbb{E}_{\theta \in \Theta} \left[ \mathbb{E} \left[ \sum_{t=0}^{\infty} \gamma^t r_t | s_0 \sim \rho_0(\theta), \mathbf{a}_0, \mathbf{m}_0 = \mathbf{a}, \mathbf{m}, \mathbf{a}_t, \mathbf{m}_t \sim \prod_{i \in \mathcal{T}_a} \pi_a(\cdot | h_t^i, b_t^i) \right] \right], \tag{7}$$

with $\mathcal{A}_a = \times_{i \in \mathcal{T}_a} \mathcal{A}_i$, the corresponding Bellman equation is then formulated as:

$$Q_*^i(s, \mathbf{a}, \mathbf{m}, b) = \max_{\pi_a(\cdot | h, b)} \mathcal{R}(s, \mathbf{a}, \theta) + \gamma \sum_{s' \in \mathcal{S}} \mathcal{P}_a(s' | s, a, \theta)$$
$$\sum_{b' \in \Delta(\Lambda)} p(b' | h') \sum_{\mathbf{a}', \mathbf{m}' \in \mathcal{A}_a} \pi_a(\mathbf{a}', \mathbf{m}' | h', b') Q_*^i(s', \mathbf{a}', \mathbf{m}', b') \tag{8}$$

This Q function is parameterized by $\xi$ and can be estimated via Temporal Difference (TD) loss through Eqn. 9.

$$\min_{\xi} \left( r_t + \gamma Q_\xi^i(s_{t+1}, \mathbf{a}_{t+1}, \mathbf{m}_{t+1}, b_{t+1}^i) - Q_\xi^i(s_t, \mathbf{a}_t, \mathbf{m}_t, b_t^i) \right)^2 \tag{9}$$

**Proposition 3.1.** Assume the belief is updated via Bayes' rule, the space of state, actions and belief are finite, updating value functions by Bellman equation converge to the optimal value $Q_*^i(s, \mathbf{a}, \mathbf{m}, b)$.

*Proof sketch.* The proof is done by combining the standard convergence proof of Q function with Bayesian belief update, and showing our Q function forms a contraction mapping. Next, applying Banach's fixed point theorem completes the proof. See full proof in Appendix. F.1.

The policy for $\pi_{\psi_1}$ and $\pi_{\psi_2}$ is updated by policy gradient, with detailed derivation in Appendix. F.2.

**Theorem 3.1.** The policy gradient theorem for the policies of adversarial agent $i$ is:

$$\nabla_{\psi_1} J(\psi_1) = \mathbb{E}_{s_t, h_t^i, b_t^i, a_t^i, m_t^i, i \in \mathcal{T}_a} \nabla \log \pi_{\psi_1}(a_t^i | h_t^i, m_t^i, b_t^i) Q_\xi^i(s_t, \mathbf{a}_t, \mathbf{m}_t, b_t^i) \tag{10}$$

$$\nabla_{\psi_2} J(\psi_2) = \mathbb{E}_{s_t, h_t^i, b_t^i, a_t^i, m_t^i, i \in \mathcal{T}_a} \nabla \left[ \left[ \sum_{j \in \mathcal{N}} \log \pi_{\psi_2}(m_t^{i,j} | h_t^i, b_t^i) \right] Q_\xi^i(s_t, \mathbf{a}_t, \mathbf{m}_t, b_t^i) \right] \tag{11}$$

## 4 EXPERIMENTS

### 4.1 EXPERIMENTAL SETUP

**Environments.** We evaluate the effectiveness of our approach on SMAC (Samvelyan et al., 2019), SMACv2 (Ellis et al., 2023) and MAgent (Zheng et al., 2018). In the original SMAC and SMACv2 setup, a group of MARL agents is trained against rule-based in-game AIs. We modify this setup to create **SMACDual** and **SMACv2Dual**, where the rule-based opponents are replaced by another group of MARL agents acting as attackers. For MAgent, we follow the standard self-play setting (Yang et al., 2018) without modification.

**Baselines.** We compare our method against two groups of baselines. The first consists of transferable MARL algorithms that directly overpower opponents, including UPDeT (Hu et al., 2021), MATTER (Qin et al., 2022), and DT2GS (Tian et al., 2023). The second is SUB-PLAY (Ma et al., 2024), which trains a non-transferable adversarial policy against MARL agents. For a fair comparison in our adversarial transfer setting, we implement both our method and SUB-PLAY using the

Table 1: Average rewards received by victim agents in StarCraft II, with varying victim number, type and policies. Our method demonstrate stronger transfer result in all 15 out of 15 tasks.

| Task | | UPDeT | MATTER | DT2GS | SUB-PLAY | Ours |
|---|---|---|---|---|---|---|
| **Transfer across tasks with varying numbers of agents ($\downarrow$).** | | | | | | |
| *Source Tasks* | *3m* | $6.04 \pm 0.00$ | $6.37 \pm 0.33$ | $6.99 \pm 0.94$ | $7.18 \pm 0.66$ | $\mathbf{1.91 \pm 0.47}$ |
| | *8m* | $10.31 \pm 0.75$ | $9.83 \pm 0.59$ | $8.63 \pm 1.12$ | $10.00 \pm 2.05$ | $\mathbf{5.23 \pm 0.58}$ |
| *Unseen Tasks* | *4m_vs_3m* | $20.00 \pm 0.00$ | $20.00 \pm 0.00$ | $20.00 \pm 0.00$ | $20.00 \pm 0.00$ | $\mathbf{2.76 \pm 0.44}$ |
| | *5m_vs_3m* | $20.00 \pm 0.00$ | $20.00 \pm 0.00$ | $20.00 \pm 0.00$ | $20.00 \pm 0.00$ | $\mathbf{3.43 \pm 0.90}$ |
| | *6m* | $12.57 \pm 0.84$ | $9.19 \pm 0.71$ | $8.40 \pm 1.06$ | $9.62 \pm 0.25$ | $\mathbf{4.23 \pm 1.65}$ |
| | *11m* | $17.63 \pm 2.27$ | $10.66 \pm 1.89$ | $9.52 \pm 1.12$ | $18.42 \pm 3.53$ | $\mathbf{6.83 \pm 1.96}$ |
| **Transfer across tasks with different agent types ($\downarrow$).** | | | | | | |
| *Source Tasks* | *Protoss_5_vs_5* | $12.18 \pm 1.88$ | $6.67 \pm 1.14$ | $8.56 \pm 2.48$ | $9.13 \pm 2.29$ | $\mathbf{2.33 \pm 0.91}$ |
| *Unseen Tasks* | *Terran_5_vs_5* | $5.96 \pm 2.16$ | $5.61 \pm 0.71$ | $4.99 \pm 1.48$ | $6.24 \pm 2.54$ | $\mathbf{3.33 \pm 0.72}$ |
| | *Zerg_5_vs_5* | $6.84 \pm 1.79$ | $7.91 \pm 1.44$ | $6.18 \pm 0.87$ | $4.43 \pm 0.67$ | $\mathbf{3.27 \pm 1.55}$ |
| **Transfer across tasks with different victim policies ($\downarrow$).** | | | | | | |
| *Source Tasks (MAPPO)* | *3m* | $7.62 \pm 0.23$ | $7.24 \pm 0.24$ | $8.33 \pm 0.74$ | $7.16 \pm 0.44$ | $\mathbf{5.64 \pm 0.13}$ |
| | *3s_vs_3z* | $10.17 \pm 0.52$ | $7.04 \pm 0.06$ | $7.39 \pm 0.16$ | $7.41 \pm 0.25$ | $\mathbf{6.53 \pm 0.20}$ |
| | *2s3z* | $13.92 \pm 1.41$ | $11.35 \pm 0.33$ | $12.86 \pm 0.99$ | $11.65 \pm 1.38$ | $\mathbf{9.28 \pm 0.30}$ |
| *Unseen Tasks (QMIX)* | *3m* | $5.93 \pm 1.11$ | $9.18 \pm 0.76$ | $8.87 \pm 1.51$ | $5.81 \pm 0.32$ | $\mathbf{3.99 \pm 0.59}$ |
| | *3s_vs_3z* | $9.92 \pm 1.56$ | $10.47 \pm 0.74$ | $11.09 \pm 2.00$ | $10.53 \pm 1.06$ | $\mathbf{7.04 \pm 0.46}$ |
| | *2s3z* | $12.36 \pm 1.66$ | $13.30 \pm 0.66$ | $11.29 \pm 1.54$ | $10.04 \pm 0.87$ | $\mathbf{9.11 \pm 1.07}$ |

UPDeT (Hu et al., 2021) backbone. All baselines share the same codebase, network architecture, and hyperparameters. Additional implementation details are provided in Appendix C.

**Evaluation Pipeline.** Our evaluation follows a two-step process. First, we train the victim policy following its original implementation in SMAC (Yu et al., 2022), SMACv2 (Ellis et al., 2023) and MAgent (Yang et al., 2018). Next, we fix the victim policy and replace its opponents with an adversarial policy. All methods are evaluated using five random seeds, with each victim and attackers sharing the same seed for consistency. In all tasks (*e.g.*, *3s_vs_3z*), the victim team always controls the agents on the left side (*i.e.*, *3s*), while attacker team controls the agents on the right side (*i.e.*, *3z*).

## 4.2 TRANSFER ATTACK IN STARCRAFT II AND MAGENT

In this section, we first evaluate the performance of transfer attack in StarCraft II, following three different transfer paradigms: 1) **Transfer across tasks with varying numbers of agents.** We train adversarial policies on *3m* and *8m*, then evaluate on more challenging tasks, *4m_vs_3m* and *5m_vs_3m*, as well as tasks with different numbers of agents, *6m* and *11m*. Extra experiments are detailed in Appendix D. 2) **Transfer across tasks with different agent types.** We train on *Protoss_5_vs_5*, and evaluate on a diverse set of tasks: *Terran_5_vs_5* and *Zerg_5_vs_5*. 3) **Transfer across victims with different victim policies.** We train adversarial policies individually against victims trained using MAPPO (Yu et al., 2021) on tasks *3m*, *3s_vs_3z*, and *2s3z*, and evaluate transfer performance against victims trained using QMIX (Rashid et al., 2018) on the same tasks. We evaluate performance in paradigm 2 on SMACv2 due to its support of uncertainty in agent types, and evaluate paradigms 1 and 3 on more controllable SMAC environment.

As shown in Table 1, our method consistently outperforms all baselines in zero-shot generalization, reducing victim rewards by an average of 22% compared with the best-performing baseline and achieving superior performance across 15 tasks with varying victim numbers, types, and policies. On source tasks, LLM-provided tactics act as external knowledge that stabilize and align attack behaviors, whereas baselines often fail to disentangle task-specific representations. On unseen tasks, our method transfers more effectively, surpassing the best baseline by +51% on scenarios with different victim numbers, +7% on victim types, and +9% on victim policies. This advantage extends to challenging cases (*4m_vs_3m* and *5m_vs_3m*), where victims cannot be defeated by simple overpowering.

**Attack different training paradigms.** Besides attacking victims trained against rule-based policies, we show our attack is also effective for policies trained via self-play. We evaluate the performance in MAgent environment in Battle task with different agent numbers. Following the same procedure, we train on *12_vs_12* and *30_vs_30*, and test on the unseen *20_vs_20*, *42_vs_42* and *64_vs_64* scenario. As shown in Table 2, our method outperforms all baselines in source tasks and remains consistently

Table 2: Average rewards received by victim agents in different MAgent tasks (↓). Our method successfully attack victims trained by self-play, and support attack with up to 64 agents.

|  | Task | UPDeT | MATTER | DT2GS | SUB-PLAY | Ours |
|---|---|---|---|---|---|---|
| *Source Tasks* | *12_vs_12* (Map size: 20) | -39.6 ± 2.0 | -36.6 ± 3.0 | -38.0 ± 3.1 | -41.6 ± 1.5 | **-42.1 ± 1.2** |
|  | *30_vs_30* (Map size: 30) | -291.6 ± 5.2 | -285.7 ± 9.6 | -283.3 ± 14.7 | -298.3 ± 6.5 | **-304.0 ± 2.4** |
| *Unseen Tasks* | *20_vs_20* (Map size: 25) | -62.4 ± 19.6 | -121.1 ± 0.5 | -70.4 ± 2.0 | -70.4 ± 2.0 | **-140.4 ± 3.7** |
|  | *42_vs_42* (Map size: 35) | -169.6 ± 5.7 | -312.4 ± 13.6 | -224.8 ± 4.1 | -128.6 ± 84.7 | **-325.1 ± 8.3** |
|  | *64_vs_64* (Map size: 40) | -128.0 ± 0.0 | 235.2 ± 4.4 | 293.6 ± 4.0 | -128.0 ± 0.1 | **-650.8 ± 12.0** |

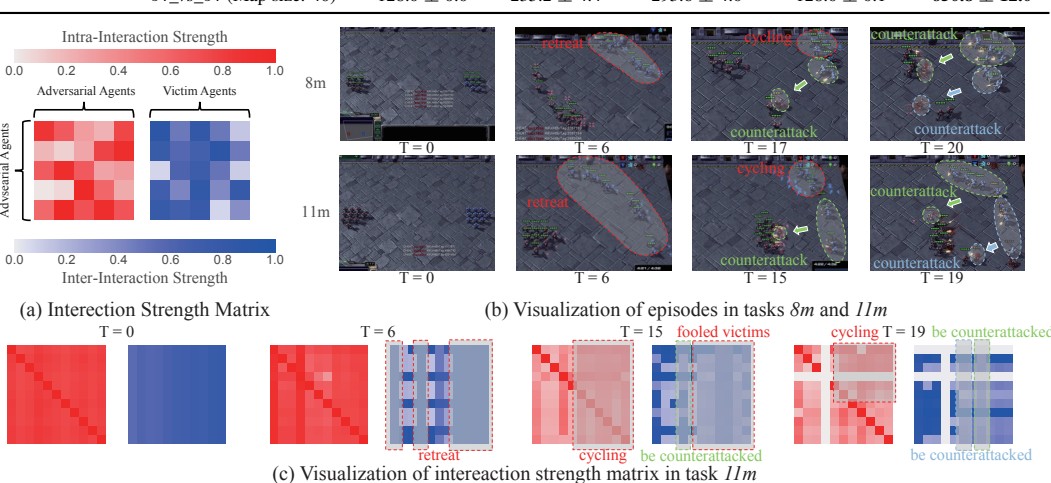

(a) Interaction Strength Matrix    (b) Visualization of episodes in tasks *8m* and *11m*

(c) Visualization of intereaction strength matrix in task *11m*

Figure 4: Analysis of Transferred Attack Behaviors. We visualize episodes from *8m* and *11m*, revealing transferred adversarial tactics and scene decomposition. We also analyze interaction strength to interpret the resulting subgame partition behavior.

effective in the unseen setting, achieving an average of 21% improvement over the best baseline. Our successful attack involving up to 64 agents further demonstrates the scalability of our method. Map sizes are noted in the table, as they define agent counts in each task.

## 4.3 DISCUSSIONS

**Countermeasures.** We investigate defense strategies by re-training against our fixed transferable attacker. As shown in Table 3, such defenses are highly effective when attackers are known *a priori*, achieving the highest reward of 20 across all tasks. However, the defense can be partially overcome by re-training the attacker against the fixed defense policy. Although the defended policy remains vulnerable to re-attacks, it demonstrates improved robust-

Table 3: Average rewards received by victim agents in StarCraft II, with varying defense methods (↓).

| Task |  | No defense | Re-train | Re-attack |
|---|---|---|---|---|
| *Source Tasks* | *3m* | 1.91 ± 0.47 | 20.00 ± 0.00 | 6.34 ± 0.96 |
|  | *8m* | 5.23 ± 0.58 | 20.00 ± 0.00 | 9.61 ± 1.41 |
| *Unseen Tasks* | *4m_vs_3m* | 2.76 ± 0.44 | 20.00 ± 0.00 | 13.47 ± 0.66 |
|  | *5m_vs_3m* | 3.43 ± 0.90 | 20.00 ± 0.00 | 17.79 ± 1.90 |
|  | *6m* | 4.23 ± 1.65 | 20.00 ± 0.00 | 9.06 ± 1.07 |
|  | *11m* | 6.83 ± 1.96 | 20.00 ± 0.00 | 10.56 ± 0.77 |

ness compared with no defense, particularly in unseen tasks such as *4m_vs_3m* and *5m_vs_3m*. We attribute this robustness to the generality of the shared weaknesses exposed by our attack: even defending against a fixed transferable attacker closes many exploitable vulnerabilities, making the policy broadly more resistant to adversaries.

**Analysis of Transferred Attack Behaviors.** In this section, we analyze how adversarial tactic acquisition and adversarial scene decomposition contribute to transfer attacks. To bridge the gap between training and testing, we select the training task *8m* and the unseen test task *11m* for analysis.

As illustrated in Figure 4(b), attackers employ coordinated tactics such as *retreat*, *counterattack*, and *circling*. In *8m*, early *retreat* disperses the victims, followed by a *counterattack* and *circling maneuver* that split and distract them, preventing effective focus fire. Victims arrive at different times and are quickly eliminated, while attacker health remains evenly distributed. Similar patterns appear

in *11m*, where attackers, despite not being trained on this scenario, again use *retreat* to scatter victims and *cycling* to attract fire, before *concentrating attacks* on isolated targets.

Scene decomposition further explains this behavior. As shown in Figure 4(c), attackers initially attend broadly to all victims, but once *retreat* begins, attention narrows to a manageable subset of agents. During *counterattack*, focus shifts to immediate targets, while *circling* agents maintain mutual awareness and distract opponents. By later stages, different groups allocate attention to specific victims in sequence, enabling coordinated elimination through localized and transferable subgames.

**Ablation Study.** We next verify the effectiveness of our components, evaluating the performance of our attack without adversarial tactic acquisition, adversarial scene decomposition and the performance on raw UPDeT (Hu et al., 2021) backbone. For our methods w/o Adversarial Tactic Acquisition, we remove the LLM-generated tactics and the Bayesian inference procedure together. Further ablations of LLM generated tactics available in paragraph below. As shown in Figure 5, our method achieves significantly higher attack capability than our ablated variants. Additionally, in difficult tasks like *4m_vs_3m*, *5m_vs_3m*, adversarial tactic acquisition and ad-

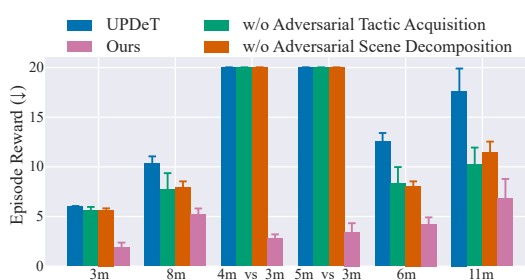

Figure 5: Ablation study on the impact of adversarial tactic acquisition and scene decomposition on zero-shot adversarial performance.

versarial scene decomposition are unable to successfully attack the task alone, and its success relies the synergy of our two attack modules.

**Ablation of LLM-generated tactics.** We next conduct further ablations on the effectiveness of LLMs for adversarial tactic generation, with experiment results available in Appendix E. First, we find not using the the adversarial tactics given by LLMs, and rely on uniform distributed tactics will significantly lower our attack performance, showing the effectiveness of LLM-generated tactics. Second, we find our attack is not highly sensitive to LLM types, and the performance remains relatively consistent with different commercial LLMs. Specifically, we tested the performance of our attack using 5 LLMs, including GPT-4o Hurst et al. (2024) which are used for other experiments, as well as Gemini 2.5 Pro (Comanici et al., 2025), GPT-o3 (OpenAI, 2025), and weaker models such as GPT-3.5 and Gemini 1.5 Flash (Reid et al., 2024). The results do not vary significant across LLMs, suggesting that the common knowledge needed to generate effective adversarial strategies is available in most commercial-level LLMs, and that highly advanced models are not a prerequisite for our attacks.

## 5 CONCLUSION

In this paper, we propose a transferable adversarial policy framework for mixed cooperative-competitive games, enabling zero-shot attacks in previously unseen scenarios. First, attackers perform adversarial tactic acquisition, iteratively extracting attack strategies that successfully deceive victims using LLMs during training, and inferring the distribution over tactics via Bayesian inference at test time. In addition, we introduce adversarial scene decomposition, which partitions attack scenarios into smaller, transferable subgames that consistently elicit adversarial behavior, based on interactions between attacker and victim teams. Alongside, we also provide a proof of the convergence of our approach. Empirically, we demonstrate that adversarial policy transfer is effective in StarCraft II and MAgent across 20 tasks, with up to 64 victim agents of varying numbers, types, and policies. Training against our attack addresses common vulnerabilities in victim policies and enhances robustness to subsequent re-attacks.

## 6 ETHICS STATEMENT

Our work reveals shared vulnerabilities across diverse MARL policies and environments. Its main positive impact is to enable efficient and accurate robustness assessment without training a separate attack for each policy. In addition, the method supports robust multi-task MARL: in settings where defending against worst-case perturbations for every task is challenging, our transferable attack

provides a fast, cost-effective proxy for such scenarios. While the approach could potentially be misused to attack MARL systems, we find defense against our attack is possible, and increase the robustness of the system. We therefore conclude that the benefits of this work outweigh possible security risks.

## 7 REPRODUCIBILITY STATEMENT

Our code is provided in the supplementary material. Detailed pseudocode for the framework is available in Appendix A, and Appendix C contains implementation details for our methods and baselines, including hyperparameters and the final LLM-generated tactics used to train adversarial policies for deployment in unseen tasks.

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

# APPENDIX FOR "ADVERSARIAL POLICY TRANSFER IN MIXED COOPERATIVE-COMPETITIVE GAMES"

**Declaration of LLM usage.** LLMs were employed for text polishing and as an integral method in this study. The authors have thoroughly reviewed and validated all content presented in the paper.

## A  PSEUDO CODE FOR OUR TRANSFERABLE ADVERSARIAL POLICY

In Section 3.1, we use a large language model (LLM) to iteratively summarize the set of identified tactics and their proportions across training scenarios, which we treat as ground-truth types for Bayesian inference. Specifically, the LLM first outputs a set of candidate tactics, then estimates their proportions for each scenario. Adversarial agents learn to infer these ground-truth tactics via Bayesian inference. We then refine both the set and proportions based on episode rewards and trajectories generated by a transferable policy trained on the previously extracted tactics. In Section 3.2, during policy execution, each agent adaptively reduces its view of the scenario by partitioning it into manageable, transferable subgames that generalize across environments, enabling more effective tactic execution. The full process is detailed in the pseudo code below.

---

**Algorithm 1** Transferable adversarial policy framework in our paper

---

Initialize prompt $P$ with successful attack trajectories for each training scenarios.
**for** $i = 1$ to $iterations\_num$ **do**
 Use $P$ as the input of LLM and get the set of all tactics $\Lambda$ and their proportions $\{\lambda_{LLM}^{env}\}_{env \in Envs}$ for each adversarial scenario.
 **for** $env \in Envs$ **do**
  **for** $episode = 1$ to $episodes\_num$ **do**
   **for** $t = 1$ to $max\_episode\_len$ **do**
    Use Equation 4 to update the beliefs over tactics based on the agents' current views.
    Use adversarial scene decomposition to select the proper actions.
   **end for**
   Use $\lambda_{LLM}^{env}$ as ground-truth type for the training scenario to update the dynamics model $P_\phi$ by Equation 5.
   Use Equation 10, 11 and 9 to update the policy for $\pi_{\psi_1}$ and $\pi_{\psi_2}$, as well as the shared critic $Q_\xi^i$.
  **end for**
 **end for**
 **for** $env \in Envs$ **do**
  Interact with $env$ to get action sequences $traj$ of every agents and total reward $r$.
  Translate $traj, r$ into textual descriptions $P_{i,env}$
  $P \leftarrow P + P_{i,env}$
 **end for**
**end for**

---

## B  DETAILS ON THE LLM PROMPTS AND OUTPUTS

Above all, we first clarify how LLMs are used to generate adversarial tactics. Specifically, we pass the actions and rewards of the agents to the LLM. To transform input trajectories into prompts, we first convert the actions into text (e.g., "Attack Enemy 0, moving north"). We then concatenate these action descriptions from each agent at each timestep, along with the associated rewards (e.g., "In step x, the actions of each agent are (...)"). While coordinates and health points may provide valuable information for adversarial tactics acquisition, we have found that including them does not necessarily improve adversarial performance. In fact, adding these details often complicates the prompts, making them harder for the LLM to process. We plan to explore this further in future work.

After that, the LLMs analyze the common patterns across different trajectories with the help of external common sense and identify the set of tactics present in the trajectories based on its rich external knowledge and common sense. For example, in the adversarial settings of Figure 1, LLMs

first identify that the whole adversarial team adopts the "Disperse" tactic. Then, LLMs observe that the team adopts the "Focus-fire" tactic. Along with other trajectories, LLMs identify the set of tactics used.

The LLMs then estimate the proportion of each tactic across different trajectories. Specifically, LLMs mark the time span of each tactic in the trajectories. For instance, LLMs analyze that in the *3m* scenario, agents adopt the "Disperse" tactic from timestep 0 to 6, followed by "Focus-fire" from timestep 6 to the end. By doing so, LLMs calculate the proportion of each tactic, which acts as the adversarial tactics for the scenario.

For example, the LLM first proposes a set of tactics "[*Focused Fire*, *Flanking Maneuver*, *Spread Attack*, *Coordinated Harassment*]", and assigns the proportion of tactics $[0.5, 0.3, 0.1, 0.1]$ to the *3m* task as its latent embedding. Similarly, the LLM assigns the proportion $[0.25, 0.25, 0.3, 0.2]$ to the *8m* task as its latent embedding. These two vectors are then used as the ground-truth latent embeddings for the corresponding environments.

We present our initial LLM prompt for SMAC in the transfer setting across tasks with varying numbers of agents, along with successful attack trajectories. Only the first three timesteps are shown for brevity.

Listing 1: Initial prompt for transfer attack in StarCraft II (varying number of agents)

```
We plan to study the transfer problem in multi-agent reinforcement
    learning on the SMAC environment.

Here's the basic situation: we have two maps, 3m and 8m. For the 3m map,
    the battle involves 3 marines versus 3 marines. For the 8m map, the
    battle involves 8 marines versus 8 marines. In addition, we have
    multi-agent models trained on these two maps. These models can
    control our units in the game and compete against computer-controlled
     enemy units on these two maps, achieving victory and obtaining the
    highest rewards.

Now, we want to train adversarial agents. These adversarial agents will
    replace the computer-controlled units and fight against the trained
    models, aiming to disrupt the actions of the trained models,
    preventing them from executing normal actions and reducing the
    rewards they obtain.

By training adversarial agents on the two maps, we hope that the trained
    agents can also be effective in attacking on other marine-based maps.
     Therefore, to achieve this attacking effect, I hope you can suggest
    strategies that the adversarial multi-agents can adopt in this
    scenario. Additionally, for the maps above, how can these strategies
    be combined? If possible, please express this quantitatively.

For example, in this scenario, there could be four strategies: A, B, C,
    and D. For the 3m map, you could use 0.1 of strategy A and 0.2 of
    strategy B. For the 8m map, you could use 0.3 of strategy C and 0.7
    of strategy D.

The strategies do not necessarily have to be four; you should list them
    according to your analysis. Additionally, you need to provide a
    reasonable explanation for the quantitative combination of strategies
     for each map.

In addition, we have trained adversarial agents as expert agents in each
    of the maps mentioned above, which replace the computer-controlled
    units and fight against the trained models, and listed their actions
    in one evaluation below. You can provide your analysis of these
    actions and behaviors, and refer to them to provide appropriate
    strategies.

For 3m_expert, in step 0, the actions of agents are (moving south, moving
     east, moving south); in step 1, the actions of agents are (moving
```

```
    south, stopping without taking any action, moving south); in step 2,
    the actions of agents are (moving south, moving south, moving south);
     in step 3...

For 8m_expert, in step 0, the actions of agents are (moving north, moving
    north, moving north, moving north, moving north, moving north,
    moving north, moving north); in step 1, the actions of agents are (
    moving north, moving north, moving north, moving north, moving north,
     moving north, moving north, moving north); in step 2, the actions of
     agents are (moving north, moving north, moving north, moving north,
    moving north, moving north, moving north, moving north); in step 3...
```

This yields the first round of tactics distilled by the LLM:

Listing 2: First round of tatics for transfer attack in StarCraft II (varying number of agents)

```
To effectively train adversarial agents in the SMAC environment that can
    generalize across different map sizes (3m, 8m), it's essential to
    design a set of robust strategies. These strategies should disrupt
    the trained multi-agent models' operations, reduce their rewards, and
     maintain effectiveness across varying scales of engagements.

Proposed Adversarial Strategies
Focused Fire (FF)

Description: All adversarial marines concentrate their attacks on a
    single target at a time to eliminate enemy units rapidly.

Objective: Quickly reduce the number of enemy units, decreasing the
    opposing team's firepower and coordination.

Flanking Maneuver (FM)

Description: Adversarial marines position themselves to attack enemy
    units from the sides or rear, exploiting positional vulnerabilities.

Objective: Disrupt the enemy's formation and force them to split their
    attention, leading to potential miscoordination.

Spread Attack (SA)

Description: Adversarial marines spread out to engage multiple enemy
    units simultaneously.

Objective: Prevent the enemy from concentrating their firepower,
    increasing the likelihood of survival for adversarial units.

Coordinated Harassment (CH)

Description: Adversarial marines employ hit-and-run tactics, constantly
    repositioning to keep the enemy off-balance.

Objective: Induce errors in the enemy's decision-making process and
    disrupt their planned actions.

Quantitative Strategy Combinations for Each Map
To tailor these strategies effectively across different map sizes, we
    assign specific weights to each strategy based on the map's scale and
     the dynamics of engagement.

1. 3m Map (3 Marines vs. 3 Marines)
Strategy Combination:

Focused Fire (FF): 50%
```

```
Flanking Maneuver (FM): 30%

Spread Attack (SA): 10%

Coordinated Harassment (CH): 10%

Rationale:

Focused Fire (50%): In small engagements, eliminating enemy units quickly
    is crucial to gaining a tactical advantage.

Flanking Maneuver (30%): With fewer units, effective flanking can
    significantly disrupt the enemy's formation.

Spread Attack & Coordinated Harassment (20% combined): Limited utility in
    very small engagements but can provide slight unpredictability.

2. 8m Map (8 Marines vs. 8 Marines)
Strategy Combination:

Focused Fire (FF): 25%

Flanking Maneuver (FM): 25%

Spread Attack (SA): 30%

Coordinated Harassment (CH): 20%

Rationale:

Spread Attack (30%): With more units, spreading out to engage multiple
    enemies prevents the opposing team from concentrating their attacks.

Focused Fire & Flanking Maneuver (25% each): Still important for
    targeting key units and disrupting formations.

Coordinated Harassment (20%): Enhanced with more units to effectively
    execute hit-and-run tactics.

Strategy Integration Across Maps
The proposed strategy weights are designed to scale logically between the
    3m and 8m maps, ensuring that the adversarial agents can adapt their
    approach based on the number of units and the map size. For instance:

Focused Fire is more heavily weighted in smaller maps where eliminating
    even a single unit can dramatically shift the balance.

Spread Attack gains prominence in larger maps where distributing attacks
    can effectively dilute the enemy's focus.

Flanking Maneuver maintains a consistent importance across maps but
    adjusts slightly based on the number of units.

Coordinated Harassment provides additional disruption in larger
    engagements but remains a supporting strategy in smaller ones.

Quantitative Summary
Strategy 3m Map (%) 8m Map (%)
Focused Fire (FF) 50 25
Flanking Maneuver (FM) 30 25
Spread Attack (SA) 10 30
Coordinated Harassment (CH) 10 20
Total 100 100
```

```
Implementation Considerations
Dynamic Strategy Adjustment:

Depending on the real-time state of the battle (e.g., remaining units,
    health levels), the adversarial agents can dynamically adjust the
    weights of these strategies to respond to changing conditions.

Learning-Based Weight Optimization:

Utilize reinforcement learning to fine-tune the strategy weights based on
    performance metrics across different maps. This can enhance
    adaptability and effectiveness beyond the initial heuristic-based
    assignments.

Cross-Map Generalization:

By training adversarial agents with a diverse set of strategy
    combinations on both 3m and 8m maps, the agents can learn to
    interpolate strategies effectively for intermediate map sizes.

Scalability:

Ensure that each strategy scales appropriately with the number of units.
    For example, Focused Fire should prioritize high-value targets
    proportionally as the number of adversarial units increases.

Conclusion
By defining clear adversarial strategies and assigning them proportionate
    weights based on map size, the adversarial agents can effectively
    disrupt trained multi-agent models across different environments. The
    proposed quantitative combinations ensure that the adversaries
    remain versatile and potent, whether operating in smaller skirmishes
    or larger battles, thereby enhancing the robustness and
    transferability of the trained adversarial agents to the target maps.
```

Next, we iteratively refine the extracted tactics based on the episode rewards and trajectories generated by a transferable adversarial policy trained using the previously proposed tactics. We encode the above information as textual input to continue the same dialogue with the LLM and obtain refined tactics:

Listing 3: Prompt used to refine tactics for transfer attack in StarCraft II (varying number of agents)

```
Here are the strategies you have provided us with, as well as the
    strategy combinations for each map.

Focused Fire (FF)
Description: All adversarial marines concentrate their attacks on a
    single target at a time to eliminate enemy units rapidly.
Objective: Quickly reduce the number of enemy units, decreasing the
    opposing team's firepower and coordination.

Flanking Maneuver (FM)
Description: Adversarial marines position themselves to attack enemy
    units from the sides or rear, exploiting positional vulnerabilities.
Objective: Disrupt the enemy's formation and force them to split their
    attention, leading to potential miscoordination.

Spread Attack (SA)
Description: Adversarial marines spread out to engage multiple enemy
    units simultaneously.
Objective: Prevent the enemy from concentrating their firepower,
    increasing the likelihood of survival for adversarial units.

Coordinated Harassment (CH)
```

```
Description: Adversarial marines employ hit-and-run tactics, constantly
    repositioning to keep the enemy off-balance.
Objective: Induce errors in the enemy's decision-making process and
    disrupt their planned actions.

3m Map:

Strategy Combination:
Focused Fire (FF): 50%
Flanking Maneuver (FM): 30%
Spread Attack (SA): 10%
Coordinated Harassment (CH): 10%

Rationale:
Focused Fire (50%): In small engagements, eliminating enemy units quickly
     is crucial to gaining a tactical advantage.
Flanking Maneuver (30%): With fewer units, effective flanking can
    significantly disrupt the enemy's formation.
Spread Attack & Coordinated Harassment (20% combined): Limited utility in
     very small engagements but can provide slight unpredictability.

8m Map:

Strategy Combination:
Focused Fire (FF): 25%
Flanking Maneuver (FM): 25%
Spread Attack (SA): 30%
Coordinated Harassment (CH): 20%

Rationale:
Spread Attack (30%): With more units, spreading out to engage multiple
    enemies prevents the opposing team from concentrating their attacks.
Focused Fire & Flanking Maneuver (25% each): Still important for
    targeting key units and disrupting formations.
Coordinated Harassment (20%): Enhanced with more units to effectively
    execute hit-and-run tactics.

We have trained the model on 3m, 8m map according to the strategy above.
    The reward for 3m is 4.79, the reward for 8m is 12.31, and the lower
    the reward, the better the effectiveness in transferable adversarial
    attacks. For these maps, we have listed their actions in one
    evaluation below.

For 3m_round1, in step 0, actions are (move south, move south, move south
    ); in step 1, actions are (move south, move south, move south); in
    step 2, actions are (move south, move south, move south); in step 3...

For 8m_round1, in step 0, actions are (move west, move south, move south,
     move south, move south, move south, move south, move west); in step
    1, actions are (move north, move south, move south, move south, move
    south, move south, move south, move north); in step 2, actions are (
    move north, move south, move south, move south, move south, move
    south, move south, move north); in step 3...

Please provide your analysis of actions above and adjust the proportions
    of the strategies for 3m, 8m to reduce the reward of each map.
```

In this paradigm, we iteratively extract tactics that successfully deceive victims using LLMs, and use these tactics as ground truth to train the dynamics model for Bayesian inference at test time.

## C  ADDITIONAL DETAILS ON EXPERIMENTS

### C.1  ADVERSARIAL TACTICS USED IN EXPERIMENTS

In Section 3.1, we emphasize that we select the models with the best adversarial performance on training tasks and deploy them to target tasks without further finetuning. And now we present the tactics used by these models:

Listing 4: Tactics used to train models for adversarial transfer

```
Starcraft II (Transfer across tasks with varying numbers of agents,
    marines series):
Strategy 3m Map (%) 8m Map (%)
Focused Fire (FF): 50 25
Flanking Maneuver (FM): 30 25
Spread Attack (SA): 10 30
Coordinated Harassment (CH): 10 20

Starcraft II (Transfer across tasks with varying numbers of agents,
    stalkers_vs_zealots series):
Strategy 3s_vs_3z Map (%) 3s_vs_5z Map (%)
Focused Aggression: 30 20
Swarming: 25 40
Flanking Maneuvers: 25 25
Disruptive Movement: 20 15

Starcraft II (Transfer across tasks with varying numbers of agents,
    combination of marines and stalkers_vs_zealots series):
Strategy 3m Map (%) 3s_vs_3z Map (%) 3s_vs_5z Map (%) 2s3z Map (%) 8m Map
    (%)
Aggressive Target-Focused Attack: 60 0 0 0 30
Divide and Conquer: 0 30 20 50 50
Kiting and Evasive Maneuvers: 0 70 80 30 0
Focus-fire Interrupt: 0 0 0 20 20

Starcraft II (Transfer across tasks with different agent types):
Strategy Protoss_5_vs_5 Map (%)
Aggressive Target-Focused Attack: 40
Harassment and Distracting Movements: 30
Defensive Stalling and Counterattacks: 0
Coordinated Ambushes or Traps: 30

Starcraft II (Transfer across tasks with different victim policies):
Strategy 3m Map (%) 3s_vs_3z Map (%) 2s3z Map (%)
Aggressive Target-Focused Attack: 60 0 0
Divide and Conquer: 0 30 50
Kiting and Evasive Maneuvers: 0 70 30
Focus-fire Interrupt: 0 0 20

MAgents:
Strategy 12_vs_12 (%) 30_vs_30 (%)
Bait: 15 20
Flank: 25 35
Focus Fire: 20 30
Disruption: 40 15
```

### C.2  IMPLEMENTATION DETAILS

Implementations of UPDeT (Hu et al., 2021), MATTER (Qin et al., 2022), DT2GS (Tian et al., 2023), and SUB-PLAY (Ma et al., 2024) share the same codebase and hyperparameters. For UPDeT (Hu et al., 2021), we perform observation decomposition $o_t^i = [o_t^{i,1}, \ldots, o_t^{i,\mathcal{N}}]$, treating the decomposed elements $\{o_t^{i,j}\}_{j \in \mathcal{N}}$ as input tokens to a self-attention encoder (Vaswani et al., 2017). And we encode the history information as an additional recurrent token. For MATTER (Qin et al., 2022), we construct

the explainer network using the same architecture as the dynamics model in our framework and train it using Equation 5, replacing $\lambda_{LLM}$ with the basis vector. For DT2GS (Tian et al., 2023), we follow the original paper's procedure to select a skill for each agent, then encode the selected skill as a token and feed it into a Transformer (Vaswani et al., 2017) to compute the action distribution. For SUB-PLAY (Ma et al., 2024), we apply random masking to a subset of agents with probability $p_u$ to implement uncertainty limitation. All baselines and our framework use the same actor network architecture.

For multi-task training in Section 4.2, we adopt the curriculum learning paradigm from (Wang et al., 2020), training the transfer policy sequentially across the training scenarios. Additionally, for iterative tactic acquisition, we prompt the LLM three times in each transfer setting and train the adversarial transfer policies from scratch in each iteration to ensure fairness. Final tactic extraction results are shown in Appendix C.1.

During training, we use shared networks for different agents to improve the generalization of the networks. This results in the training of four networks: the dynamics network $P_\phi$, the scene decomposition network $\pi_{\psi_2}$, the action selection network $\pi_{\psi_1}$, and the shared critic network $Q_\xi$. Additionally, both $\pi_{\psi_1}$ and $\pi_{\psi_2}$ share the same self-attention encoder, which processes the observations. However, $\pi_{\psi_1}$ has a dedicated action head.

For training stability, the additional networks are designed to assist the attack process. Initially, the policies and additional networks output random actions to focus on exploration. As training progresses, the information from the additional networks becomes increasingly helpful in guiding the attack policy. Empirically, we observe that the results of our attack are consistent over five random seeds.

In terms of computational overhead, the algorithms use 17M of video memory and take about one day to train the most complex setting on a single 3090 GPU. The training time for our framework is approximately 60% longer than that of the simplest method, UPDeT.

We present all hyperparameters of each environment in the table below:

Table 4: Hyperparameters in SMACDual environment.

| Hyperparameter | Value | Hyperparameter | Value | Hyperparameter | Value |
|---|---|---|---|---|---|
| rollouts | 20 | mini-batch num | 1 | PPO epoch | 5 |
| gamma | 0.99 | max grad norm | 10 | PPO clip | 0.05 |
| gain | 0.01 | max episode len | 200 | entropy coef | 0.01 |
| Transformer depth | 2 | actor lr | 5e-4 | eval episode | 32 |
| hidden dim | 128 | critic lr | 5e-4 | optimizer | Adam |
| Huber loss | True | Transformer head | 3 | GAE lambda | 0.95 |
| use PopArt | True | belief lr | 5e-4 | Huber delta | 10 |
| total timestep | 1000000 | belief epoch | 5 | critic epoch | 5 |
| SUB-PLAY $p_u$ | 0.5 | MATTER few-shot Episodes | 40 | DT2GS $n_k$ | 4 |

Table 5: Hyperparameters in MAgent environment.

| Hyperparameter | Value | Hyperparameter | Value | Hyperparameter | Value |
|---|---|---|---|---|---|
| rollouts | 1 | mini-batch num | 1 | PPO epoch | 5 |
| gamma | 0.99 | max grad norm | 10 | PPO clip | 0.05 |
| gain | 0.01 | max episode len | 400 | entropy coef | 0.01 |
| Transformer depth | 2 | actor lr | 5e-4 | eval episode | 32 |
| hidden dim | 128 | critic lr | 5e-4 | optimizer | Adam |
| Huber loss | True | Transformer head | 3 | GAE lambda | 0.95 |
| use PopArt | True | belief lr | 5e-4 | Huber delta | 10 |
| total timestep | 800000 | belief epoch | 5 | critic epoch | 5 |
| SUB-PLAY $p_u$ | 0.5 | MATTER few-shot Episodes | 40 | DT2GS $n_k$ | 4 |

# D ADDITIONAL EXPERIMENTAL RESULTS IN STARCRAFT II

Table 6: Average rewards received by victim agents in different SMAC *stalkers_vs_zealots* series tasks (↓). Our method demonstrate stronger transfer result in 4 out of 5 tasks.

| Task | | UPDeT | MATTER | DT2GS | SUB-PLAY | Ours |
|------|--|-------|--------|-------|----------|------|
| *Source Tasks* | *3s_vs_3z* | $16.34 \pm 0.24$ | $8.96 \pm 0.63$ | $13.88 \pm 0.34$ | $17.98 \pm 4.51$ | **7.53 ± 0.20** |
| | *3s_vs_5z* | $14.42 \pm 2.67$ | $10.45 \pm 1.27$ | **3.2 ± 0.01** | $9.25 \pm 0.93$ | **3.2 ± 0.04** |
| *Unseen Tasks* | *3s_vs_4z* | $15.44 \pm 0.71$ | $11.89 \pm 0.56$ | $16.76 \pm 0.26$ | $14.79 \pm 0.64$ | **6.82 ± 1.50** |
| | *4s_vs_3z* | $14.85 \pm 1.19$ | $8.88 \pm 0.96$ | $15.88 \pm 2.39$ | $13.46 \pm 3.72$ | **6.82 ± 0.53** |
| | *4s_vs_4z* | $14.91 \pm 1.03$ | $11.70 \pm 0.89$ | $10.29 \pm 3.00$ | $16.25 \pm 1.20$ | **8.03 ± 0.79** |

In Section 4.2, we evaluate transfer attack performance in SMAC under three different transfer paradigms, one of which involves transferring across tasks with varying numbers of agents. We primarily assess this using the *marines* series. In addition, we present results on the *stalkers_vs_zealots* series to demonstrate that our method can also support other agent types within this transfer paradigm. Specifically, we train adversarial policies on *3s_vs_3z* and *3s_vs_5z*, and then evaluate on tasks *3s_vs_4z*, *4s_vs_3z* and *4s_vs_4z*. As shown in Table 6, our method demonstrate stronger transfer result in 4 out of 5 tasks and remains consistently effective in the unseen setting, achieving an average of 16% improvement over the best baseline.

Table 7: Average rewards received by victim agents in different SMAC *stalkers_vs_zealots* series tasks (↓). Our method demonstrate stronger transfer result in 4 out of 5 tasks.

| Task | | UPDeT | MATTER | DT2GS | SUB-PLAY | Ours |
|------|--|-------|--------|-------|----------|------|
| *Source Tasks* | *3m* | $10.13 \pm 5.75$ | $15.60 \pm 1.53$ | $11.84 \pm 4.06$ | $6.40 \pm 0.54$ | **5.97 ± 0.27** |
| | *3s_vs_3z* | $7.26 \pm 0.65$ | $7.36 \pm 0.07$ | $7.22 \pm 0.40$ | $7.41 \pm 0.18$ | **6.53 ± 0.19** |
| | *3s_vs_5z* | $3.53 \pm 0.35$ | $3.17 \pm 0.08$ | $3.48 \pm 0.32$ | $3.87 \pm 0.43$ | **2.86 ± 0.17** |
| | *2s3z* | $14.21 \pm 1.57$ | $15.81 \pm 0.86$ | $12.26 \pm 1.18$ | $12.53 \pm 1.52$ | **10.30 ± 1.22** |
| | *8m* | $11.82 \pm 0.82$ | $15.03 \pm 0.38$ | $11.07 \pm 1.19$ | $9.50 \pm 0.59$ | **9.09 ± 0.99** |
| *Unseen Tasks* | *2m_vs_1z* | $5.06 \pm 0.15$ | $5.03 \pm 0.05$ | $5.49 \pm 0.93$ | $5.06 \pm 0.15$ | **4.11 ± 0.75** |
| | *3s_vs_4z* | $5.17 \pm 0.37$ | $4.74 \pm 0.14$ | $5.42 \pm 0.62$ | $5.32 \pm 0.18$ | **4.57 ± 0.21** |
| | *4s_vs_4z* | $8.50 \pm 0.54$ | $10.03 \pm 0.59$ | $8.26 \pm 0.44$ | $7.86 \pm 0.22$ | **7.48 ± 0.31** |
| | *4s_vs_3z* | $15.67 \pm 3.55$ | $16.27 \pm 0.90$ | $14.19 \pm 3.37$ | $12.22 \pm 0.73$ | **11.14 ± 0.48** |
| | *6m* | $14.40 \pm 4.85$ | $15.90 \pm 0.76$ | $14.20 \pm 5.29$ | $11.03 \pm 5.01$ | **9.60 ± 1.51** |
| | *11m* | $16.20 \pm 2.89$ | $20.00 \pm 0.00$ | $20.00 \pm 0.00$ | $12.12 \pm 1.32$ | **10.81 ± 1.61** |
| | *3s5z* | $14.11 \pm 0.64$ | $14.09 \pm 0.34$ | $12.23 \pm 1.01$ | $14.39 \pm 0.34$ | **11.91 ± 0.66** |

In addition, we conduct experiments on a more complex series involving multiple agent types on each side. This series can be viewed as a merge of the *marines* and *stalkers_vs_zealots* benchmarks, along with several environments that contain agent types from both (*e.g.*, *2m_vs_1z*, *2s3z*, *3s5z*). As shown in Table 7, our method consistently achieves stronger transfer performance across all 11 tasks and remains effective in unseen settings, yielding an average improvement of 4% over the best baseline.

# E IMPACT OF ADVERSARIAL TACTICS

In Section 3.1, we adopt GPT-4o (Hurst et al., 2024) to generate adversarial tactics. To assess how tactic-generation methods affect adversarial transfer, we conducted additional experiments. First, we applied pattern mining by deriving latent embeddings for each task from tokenized agent types and numbers, then mining interaction patterns from the interaction, similar to the process in ODIS (Zhang et al., 2023), with the embeddings as the additional conditional input. Second, we evaluated different LLMs, including Gemini 2.5 Pro (Comanici et al., 2025), GPT-o3 (OpenAI, 2025), and weaker models such as GPT-3.5 and Gemini 1.5 Flash (Reid et al., 2024). Finally, to test robustness to hallucinations, we assigned uniform tactic proportions to each agent (e.g., $[0.25, 0.25, 0.25, 0.25]$), simulating the effect of unreliable outputs. Results are reported in Table 8.

Table 8: Average rewards received by victim agents in StarCraft II, with varying methods of generating adversarial tactics.

| Task | | Uniform Proportion | Pattern mining | Gemini 2.5 Pro | GPT-o3 | Gemini 1.5 Flash | GPT-3.5 | GPT-4o |
|------|------|------|------|------|------|------|------|------|
| *Source Tasks* | *3m* | $7.26 \pm 0.45$ | $2.24 \pm 0.08$ | $1.64 \pm 0.33$ | $2.84 \pm 0.56$ | $2.17 \pm 0.18$ | $3.74 \pm 0.67$ | $1.91 \pm 0.47$ |
| | *8m* | $12.81 \pm 0.49$ | $7.39 \pm 0.07$ | $6.00 \pm 0.28$ | $6.00 \pm 0.28$ | $6.21 \pm 0.29$ | $6.54 \pm 0.71$ | $5.23 \pm 0.58$ |
| *Unseen Tasks* | *4m_vs_3m* | $20.00 \pm 0.00$ | $11.76 \pm 0.72$ | $2.10 \pm 0.44$ | $2.96 \pm 0.23$ | $2.50 \pm 0.55$ | $6.17 \pm 0.92$ | $2.76 \pm 0.44$ |
| | *5m_vs_3m* | $20.00 \pm 0.00$ | $20.00 \pm 0.00$ | $3.13 \pm 0.74$ | $3.24 \pm 0.30$ | $3.64 \pm 0.48$ | $17.60 \pm 5.37$ | $3.43 \pm 0.90$ |
| | *6m* | $11.40 \pm 0.39$ | $7.06 \pm 0.16$ | $4.39 \pm 0.48$ | $4.38 \pm 0.61$ | $4.71 \pm 1.12$ | $7.71 \pm 0.72$ | $4.23 \pm 1.65$ |
| | *11m* | $13.85 \pm 0.23$ | $12.88 \pm 0.21$ | $5.25 \pm 0.41$ | $5.92 \pm 1.06$ | $6.49 \pm 0.61$ | $7.34 \pm 0.57$ | $6.83 \pm 1.96$ |

These results yield three main conclusions. First, tactics generated by powerful LLMs (e.g., Gemini 2.5 Pro, GPT-4o) consistently outperform pattern mining, as LLMs leverage external knowledge and common-sense reasoning (*e.g.*, from SC2 tutorials and online forums) to propose more sophisticated adversarial strategies, whereas pattern mining is limited to interaction data. Second, the choice of LLM has little effect once the model is sufficiently strong (e.g., Gemini 1.5 Flash, Gemini 2.5 Pro, GPT-o3, GPT-4o), with only minor scenario-specific differences, suggesting that effective tactics in cooperative–competitive games largely depend on common sense. In contrast, weaker models (e.g., GPT-3.5) degrade performance, showing that our method is not fully immune to poor tactic generation. Third, uniform tactic proportions consistently underperform relative to LLM outputs, indicating some sensitivity to hallucinations. Nevertheless, repeated prompting produced stable outputs, and the generated tactics (*e.g.*, for *3m*: *Flanking Maneuver*, *Coordinated Harassment*, followed by *Spread Attack* and *Focused Fire*) closely matched observed behaviors (Fig. 1), suggesting that hallucinations are not a significant concern in practice.

# F    PROOFS

## F.1    PROOF OF PROPOSITION 3.1

Here we present the full proof of convergence of $Q^i(s, \mathbf{a}, \mathbf{m}, b)$. We first show that updating $Q^i$ by Bellman operator $\mathcal{B}$ is a contraction on Banach space, with $\mathcal{B}$ defined as:

$$\left(\mathcal{B}Q^i\right)(s, \mathbf{a}, \mathbf{m}, b) = r_t + \gamma \sum_{s' \in \mathcal{S}} \mathcal{P}_a(s'|s, a, \theta)$$
$$\sum_{b' \in \Delta(\Lambda)} p(b'|h') \sum_{\mathbf{a}', \mathbf{m}' \in \mathcal{A}_a} \pi_a(\mathbf{a}', \mathbf{m}'|h', b') Q_*^i(s', \mathbf{a}', \mathbf{m}', b'). \quad (12)$$

Define two Q functions $Q_1^i(s, \mathbf{a}, \mathbf{m}, b)$ and $Q_2^i(s, \mathbf{a}, \mathbf{m}, b)$, we need to show the Bellman operator $\mathcal{B}$ is a contraction in sup-norm:

$$
\begin{aligned}
&||\mathcal{B}Q_1^i - \mathcal{B}Q_2^i||_\infty \\
&= \max_{s,\mathbf{a},\mathbf{m},b} \left| \left(\mathcal{B}Q_1^i\right)(s, \mathbf{a}, \mathbf{m}, b) - \left(\mathcal{B}Q_2^i\right)(s, \mathbf{a}, \mathbf{m}, b) \right| \\
&= \max_{s,\mathbf{a},\mathbf{m},b} \left| r_t + \gamma \sum_{s' \in \mathcal{S}} \mathcal{P}_a(s'|s, a, \theta) \sum_{b' \in \Delta(\Lambda)} p(b'|h') \sum_{\mathbf{a}',\mathbf{m}' \in \mathcal{A}_a} \pi_a(\mathbf{a}', \mathbf{m}'|h', b') Q_1^i(s', \mathbf{a}', \mathbf{m}', b') \right. \\
&\quad \left. - r_t - \gamma \sum_{s' \in \mathcal{S}} \mathcal{P}_a(s'|s, a, \theta) \sum_{b' \in \Delta(\Lambda)} p(b'|h') \sum_{\mathbf{a}',\mathbf{m}' \in \mathcal{A}_a} \pi_a(\mathbf{a}', \mathbf{m}'|h', b') Q_2^i(s', \mathbf{a}', \mathbf{m}', b') \right| \\
&= \max_{s,\mathbf{a},\mathbf{m},b} \gamma \left| \sum_{s' \in \mathcal{S}} \mathcal{P}_a(s'|s, a, \theta) \sum_{b' \in \Delta(\Lambda)} p(b'|h') \sum_{\mathbf{a}',\mathbf{m}' \in \mathcal{A}_a} \pi_a(\mathbf{a}', \mathbf{m}'|h', b') \right. \\
&\quad \left. (Q_1^i(s', \mathbf{a}', \mathbf{m}', b') - Q_2^i(s', \mathbf{a}', \mathbf{m}', b')) \right| \\
&\leq \max_{s,\mathbf{a},\mathbf{m},b} \gamma \sum_{s' \in \mathcal{S}} \mathcal{P}_a(s'|s, a, \theta) \sum_{b' \in \Delta(\Lambda)} p(b'|h') \sum_{\mathbf{a}',\mathbf{m}' \in \mathcal{A}_a} \pi_a(\mathbf{a}', \mathbf{m}'|h', b') \\
&\quad \left| Q_1^i(s', \mathbf{a}', \mathbf{m}', b') - Q_2^i(s', \mathbf{a}', \mathbf{m}', b') \right| \\
&\leq \gamma ||Q_1^i(s', \mathbf{a}', \mathbf{m}', b') - Q_2^i(s', \mathbf{a}', \mathbf{m}', b')||_\infty
\end{aligned}
$$

$$(13)$$

Thus, $\mathcal{B}$ is a contraction operator. Finally, by Banach's fixed point theorem, with finite joint action space $\mathcal{A}$, state space $\mathcal{S}$, and assume each state-action pair is visited infinitesimally often, updating $Q^i(s_t, \mathbf{a}, \mathbf{m}, b)$ by Bellman operator $\mathcal{B}$ will converge to the optimal value function $Q^{i,*}(s_t, \mathbf{a}, \mathbf{m}, b)$. Note that the guaranteed convergence happens in tabular case. This motivates us to use MAPPO algorithm as a practical solver of this problem.

## F.2 PROOF OF THEOREM 3.1

We first discuss the policy gradient with $\pi_a(\mathbf{a}, \mathbf{m}|h, b)$.

$$\nabla_{\psi_1^i, \psi_2^i} V^i(s)$$

$$= \nabla_{\psi_1^i, \psi_2^i} \left[ \sum_{b \in \Delta(\Theta)} p(b|h) \sum_{\mathbf{a}, \mathbf{m} \in \mathcal{A}_a} \pi_a(\mathbf{a}, \mathbf{m}|h, b) Q^i(s, \mathbf{a}, \mathbf{m}, b^i) \right]$$

$$= \sum_{b \in \Delta(\Theta)} p(b|h) \sum_{\mathbf{a}, \mathbf{m} \in \mathcal{A}_a} \left[ \nabla_{\psi_1^i, \psi_2^i} \pi_a^i(a^i, m^i|h^i, b^i) Q^i(s, \mathbf{a}, \mathbf{m}, b^i) + \pi_a(\mathbf{a}, \mathbf{m}|h, b^i) \nabla_{\psi_1, \psi_2} Q^i(s, \mathbf{a}, \mathbf{m}, b) \right]$$

$$= \sum_{b \in \Delta(\Theta)} p(b|h) \sum_{\mathbf{a}, \mathbf{m} \in \mathcal{A}_a} \left[ \nabla_{\psi_1^i, \psi_2^i} \pi_a^i(a^i, m^i|h^i, b^i) Q^i(s, \mathbf{a}, \mathbf{m}, b^i) + \pi_a(\mathbf{a}, \mathbf{m}|h, b^i) \nabla_{\psi_1, \psi_2} (\mathcal{R}(s, \mathbf{a}, \theta) \right.$$

$$\left. + \gamma \sum_{s' \in \mathcal{S}} \mathcal{P}_a(s'|s, \mathbf{a}, \theta) \sum_{b' \in \Delta(\Lambda)} p(b'|h') \sum_{\mathbf{a}', \mathbf{m}' \in \mathcal{A}_a} \pi_a(\mathbf{a}', \mathbf{m}'|h', b') Q^i(s', \mathbf{a}', \mathbf{m}', b')) \right]$$

$$= \sum_{b \in \Delta(\Theta)} p(b|h) \sum_{\mathbf{a}, \mathbf{m} \in \mathcal{A}_a} \left[ \nabla_{\psi_1^i, \psi_2^i} \pi_a^i(a^i, m^i|h^i, b^i) Q^i(s, \mathbf{a}, \mathbf{m}, b^i) + \gamma \pi_a(\mathbf{a}, \mathbf{m}|h, b^i) \sum_{s' \in \mathcal{S}} \mathcal{P}_a(s'|s, \mathbf{a}, \theta) \right.$$

$$\left. \sum_{b' \in \Delta(\Lambda)} p(b'|h') \nabla_{\psi_1, \psi_2} \left[ \sum_{\mathbf{a}', \mathbf{m}' \in \mathcal{A}_a} \pi_a(\mathbf{a}', \mathbf{m}'|h', b') Q^i(s', \mathbf{a}', \mathbf{m}', b') \right] \right]$$

$$= \sum_{s' \in \mathcal{S}} \sum_{t=0}^{\infty} Pr(s \to s', t, \pi_a) \sum_{b \in \Delta(\Theta)} p(b|h) \sum_{\mathbf{a}, \mathbf{m} \in \mathcal{A}_a} \left[ \nabla_{\psi_1^i, \psi_2^i} \pi_a^i(a^i, m^i|h^i, b^i) Q^i(s', \mathbf{a}, \mathbf{m}, b^i) \right]$$

$$\tag{14}$$

Then we have:

$$\nabla_{\psi_1^i, \psi_2^i} J^i(\psi_1^i, \psi_2^i)$$

$$= \nabla_{\psi_1^i, \psi_2^i} V^i(s)$$

$$= \sum_{s \in \mathcal{S}} \sum_{t=0}^{\infty} Pr(s \to s, t, \pi_a) \sum_{b \in \Delta(\Theta)} p(b|h) \sum_{\mathbf{a}, \mathbf{m} \in \mathcal{A}_a} \left[ \nabla_{\psi_1^i, \psi_2^i} \pi_a^i(a^i, m^i|h^i, b^i) Q^i(s, \mathbf{a}, \mathbf{m}, b^i) \right] \tag{15}$$

$$\propto \sum_{s \in \mathcal{S}} d^{\pi_a}(s) \sum_{b \in \Delta(\Theta)} p(b|h) \sum_{\mathbf{a}, \mathbf{m} \in \mathcal{A}_a} \left[ \nabla_{\psi_1^i, \psi_2^i} \pi_a^i(a^i, m^i|h^i, b^i) Q^i(s, \mathbf{a}, \mathbf{m}, b^i) \right]$$

$$= \mathbb{E}_{s \sim d^{\pi_a}, b \sim p(\cdot|h), \mathbf{a}, \mathbf{m} \sim \pi_a(\cdot|h, b)} \left[ \nabla_{\psi_1^i, \psi_2^i} \log \pi_a^i(a^i, m^i|h^i, b^i) Q^i(s, \mathbf{a}, \mathbf{m}, b^i) \right]$$

Then we divide $\pi_a$ into the inference that consists of 2 stages, and get:

$$\nabla_{\psi_1^i, \psi_2^i} \log \pi_a^i(a^i, m^i|h^i, b^i) = (\nabla_{\psi_1^i} \log \pi_{\psi_1^i}(a^i|h^i, m^i, b^i), \nabla_{\psi_2^i} \sum_{j \in \mathcal{N}} \log \pi_{\psi_2^i}(m^{i,j}|h^i, b^i)) \tag{16}$$

Considering the shared parameters between different agents, finally we get:

$$\nabla_{\psi_1} J(\psi_1) = \mathbb{E}_{s \sim d^{\pi_a}, b \sim p(\cdot|h), \mathbf{a}, \mathbf{m} \sim \pi_a(\cdot|h, b), i \in \mathcal{T}_a} \nabla_{\psi_1} \log \pi_{\psi_1}(a^i|h^i, m^i, b^i) Q^i(s, \mathbf{a}, \mathbf{m}, b^i) \tag{17}$$

$$\nabla_{\psi_2} J(\psi_2) = \mathbb{E}_{s \sim d^{\pi_a}, b \sim p(\cdot|h), \mathbf{a}, \mathbf{m} \sim \pi_a(\cdot|h, b), i \in \mathcal{T}_a} \sum_{j \in \mathcal{N}} \nabla_{\psi_2} \log \pi_{\psi_2}(m^{i,j}|h^i, b^i) Q^i(s, \mathbf{a}, \mathbf{m}, b^i) \tag{18}$$

