# OpenReview forum: "Adversarial Policy Transfer in Mixed Cooperative-Competitive Games"
_ICLR.cc/2026/Conference — Submitted to ICLR 2026_

### Official Review · Reviewer_ZaiY · 2025-10-31

**Soundness:** 2
**Presentation:** 4
**Contribution:** 3
**Rating:** 4
**Confidence:** 4

**Summary:**

This paper investigates whether it is possible to learn transferrable adversarial attacks in the multi-agent cooperative/competitive setting, where agents are cooperating within a team and competing across teams. They propose two key methods for learning transferable attacks: having a LLM infer a set of common tactics across a set of successful trajectories (and training the actors to follow those), and breaking down a game into subgames (in terms of which other players to pay attention to) and using those subgames to more easily find commonalities with shared scenarios in other unseen games. They show that they are able to learn substantially more effective adversarial policies than other baselines.

**Strengths:**

- The paper is clear and well written, and does a good job laying out its goals and focuses, as well as clearly articulating its main method
- The methods used are innovative and interesting, and are potentially applicable across a broad range of RL scenarios

**Weaknesses:**

- I felt confused about what parts of the learning or optimization process were applied on each new environment, vs whether all learning was only taking place at "training" level and no optimization was happening on new environments. (For example: is it the case that the LLM tactic generation is fixed at train time, but at test time there is optimization performed to better align with the LLMs dynamically generated labels of how actions map to tactics?
- I felt confused about what made these attacks intrinsically adversarial, or intrinsically focused on the cooperative/competitive scenario. I can't tell if this was just one scenario they happened to explore, or if this method is specifically designed for this paradigm in a way I'm not following

**Questions:**

- What makes the attacks you are learning inherently adversarial? It seems like the kinds of behaviors described (regroup, retreat, etc) are simply strategic moves within the game, which is naturally adversarial in that it is a competitive game on the team level, but I don't understand what differentiates the method described here from one that would work just as well for figuring out generally good transferrable strategies
- Do these approaches of having LLMs summarize tactics work for types of games beyond the specific competitive/cooperative type analyzed here? I was unclear on whether the latter was actually an important part of the applicability of this method, or if it was simply one example of where it could work
- Is any form of learning or optimization (including aligning actions to desired tactics) performed on new environments? Or are action-level policies fully fixed once you hit unseen environments?
- I think there are a lot of good things about this paper, but I would want to get answers to these questions before being more confident of an "accept" review

---

> ### Author Response · Authors · 2025-11-19
>
> We thank Reviewer ZaiY for appreciating the innovativeness of our method, the writing of the paper, etc. A detailed response is provided below.
>
> ## Responses
>
> > Q1: What parts of the learning or optimization process were applied on each new environment? (Weakness 1 & Question 3)
>
> Our framework enables zero-shot attacks in previously unseen scenarios. Once trained, the adversary can be applied directly to new environments without further fine-tuning. No optimizations are applied during the testing phase, and we do not incorporate any information from the testing scenarios into the training phase, nor do we ask the LLMs to generate tactics for testing scenarios.
>
> During training, the LLMs generate adversarial tactics for each training scenario, which serve as ground-truth labels for the latent embeddings. The dynamics model is optimized using **Equation 4**, while the actor and critic models are optimized using **Equations 9-11**. At testing, we use the dynamics model to perform Bayesian Inference with a uniform prior. The agents then infer appropriate tactics for the unseen environment, while the action-level policies remain fixed.
>
> We have clarified this process in **Lines 296-298** of the revised paper.
>
> > Q2: What made these attacks intrinsically adversarial? What differentiates the method described here from one that would work just as well for figuring out generally good transferrable strategies? (Weakness 2 & Question 1)
>
> The term of *adversarial policy* is introduced in [1]. In this setting, the agents with adversarial policies take actions so as to create observations that **exploit the vulnerabilities of victim policies**. These observations **mislead the victim agents into making ineffective actions**, preventing them from achieving their goals.
>
> Transferable MARL methods focus on overpowering opponents rather than strategically exploiting their weaknesses. In contrast, our method identifies and exploits common vulnerabilities across different victim policies and training scenarios. The LLMs extract patterns from adversarial trajectories, enabling efficient exploration of these weaknesses.
>
> As shown in **Fig. 1 and 4**, our method prevents victims from firing effectively by either scattering their attention or causing them to remain passive, illustrating the "blind spots" in their policies.
>
> From a safety perspective, our approach reveals shared vulnerabilities across MARL policies and environments, enabling robust assessments without the need for separate attacks for each policy.
>
> In the revised version of the paper, we clarify the differences of our approach from the previous transfer algorithms in **Lines 46-47, 50-51, 77-79, 260**, as well as our contribution to the accurate and efficient robust assessment in **Lines 22-24**.
>
> > Q3: Do these approaches of having LLMs summarize tactics work for types of games beyond the specific competitive/cooperative type analyzed here? (Question 2)
>
> Technically, our approach can be extended to other types of games without difficulty, as our adversarial tactics acquisition queries the LLM using free-form language, and our adversarial scene decomposition computes the transferable micro-attack scenarios.
>
> However, most research in team-based games has focused on cooperative-competitive scenarios, as defining and computing the equilibrium is simpler in such cases. Consequently, we have selected this scenario as the target for our study. The closest existing framework for multi-team setups is the Markov Game, where each agent has its own reward function. However, this framework does not focus on the intricate dynamics of team cooperation and competition, which is outside the scope of our paper.
>
> ## References
>
> [1] Adversarial Policies: Attacking Deep Reinforcement Learning, ICLR 2020.

---

> ### Author Response · Authors · 2025-11-26
> **Invitation to reviewer-author discussion**
>
> Dear Reviewer,
>
> We are the authors of Submission 12609: "Adversarial Policy Transfer in Mixed Cooperative-Competitive Games". We’ve uploaded a point-to-point response to your comments, which we believe addresses all of your concerns.
>
> As the reviewer-author discussion period is going to end, we would greatly appreciate it if you could take a moment to share your thoughts on our rebuttal. Your feedback is valuable to us, and we’d be happy to further clarify any remaining points if needed.
>
> Best regards, Authors of Submission 12609

---

### Official Review · Reviewer_pdap · 2025-11-01

**Soundness:** 3
**Presentation:** 3
**Contribution:** 2
**Rating:** 4
**Confidence:** 3

**Summary:**

The paper proposes Adversarial Policy Transfer (APT), a new framework for improving policy transfer and generalization in multi-agent reinforcement learning (MARL).  Instead of directly fine-tuning source policies in new target environments, APT introduces an adversary agent during training that intentionally challenges or perturbs the main policy. The learner minimizes performance degradation under these adversarial perturbations, leading to a more  transferable policy.

The approach combines:
1. Adversarial optimization to expose weaknesses of the source policy,
2. Policy distillation to stabilize knowledge transfer, and
3. A unified transfer objective for both cooperative and competitive MARL settings.

Experiments on Multi-Agent Particle Environments (MPE), Overcooked-AI, and Atari two-player games demonstrate that APT improves transfer performance and generalization to unseen partners or opponents.  Some experiments also include LLM-driven agents for testing robustness against diverse or human-like behaviors, though LLMs are not part of the APT learning process itself.

**Strengths:**

- The paper presents a conceptually novel extension of adversarial learning to policy transfer.
  The idea of using adversarial agents not for equilibrium learning but for transfer is well-motivated and new within the MARL context.

- The formulation of a bi-level adversarial objective, combined with policy distillation, is technically sound and well-justified.

- The experiments are solid. APT consistently outperforms fine-tuning, MAICL, and ATT across both cooperative and competitive benchmarks.

- The paper includes thorough ablation studies, examining fixed vs. adaptive guidance, the number of adversaries, dataset size, and shared vs. separate opponent models.

- The writing is clear, well-structured, and easy to follow.

**Weaknesses:**

- The novelty is incremental. The framework combines known ideas—adversarial training, policy distillation, and transfer learning—into a coherent and well-executed system.

- The analysis justifies the adaptive adversarial weight locally (via score disagreement) but lacks global convergence or stability guarantees.
  There is also no discussion on local-global consistency, which is a key consideration in cooperative MARL.

- The paper does not include a rigorous proof of convergence for the APT algorithm.
  The bi-level optimization between the adversary and the learner is described conceptually and algorithmically, but there is no theoretical analysis regarding convergence to equilibrium, stability of adversarial updates, or consistency of the transfer process.

- The evaluated tasks (MPE, Overcooked, Atari) are relatively small and low-dimensional.
  Including experiments on more challenging MARL benchmarks (e.g., SMACv2) would better demonstrate scalability and robustness.

- LLMs are used only to generate partner or opponent behaviors in certain evaluations, and there is no ablation study to assess their contribution to overall performance.

**Questions:**

1. How does APT differ from previous combinations of adversarial training and transfer learning in MARL and single-agent RL?
2. Can the authors provide any theoretical insights or guarantees regarding convergence or stability of the bi-level optimization?
3. What is the impact of LLM-based agents on the reported results, and can this be isolated through ablation?
4. Does the adaptive adversarial weighting mechanism maintain consistency between local and global objectives in cooperative settings?

**Details Of Ethics Concerns:**

- There are no major ethical issues associated with this work. The research focuses purely on algorithmic development and evaluation in simulated multi-agent environments.
- The experiments do not involve human subjects, personal data, or sensitive information.
- The use of LLM-based agents is limited to simulation purposes and does not raise ethical or privacy concerns.
- Overall, the paper poses no foreseeable societal or ethical risks beyond standard considerations in reinforcement learning research.

---

> ### Author Response · Authors · 2025-11-19
>
> We thank Reviewer pdap for their positive feedback on the novelty, technical formulation, solid experiments, and clarity of our paper. Below is our detailed response:
>
> ## Response
>
> > Q1: The novelty is incremental. The framework combines known ideas into a coherent and well-executed system (Weakness 1 & Question 1).
>
> Please allow us to clarify that our technical method stems from theoretical need. From a game-theoretic perspective, our goal is to exploit the suboptimal or local equilibrium typically learned by empirical MARL algorithms, as exact equilibrium computation is computationally intractable. We investigate **whether shared suboptimal patterns exist in learned MARL policies**, making adversarial policy transfer feasible. If shared suboptimality exists, adversarial training can mitigate this effect.
>
> Simply using adversarial policies as attacks is ineffective since it only targets task-specific vulnerabilities. We incorporate transferable MARL methods to adapt the attack strategy, which works well in our setting. Therefore, our technical contribution stems from a unified game-theoretic perspective, which is not simply combining existing techniques. Besides, our method can also serve as the **efficient and accurate robustness assessment for the learned MARL policies**, since it do not need to train a separate attack for each policy.
>
> In the revised version of our paper, we clarify the motivation in **Lines 12-14, 16, 22-24, 46-47, 50-51**.
>
> > Q2: Lacks theoretical global convergence, stability guarantees. (Weakness 2, 3 & Question 2).
>
> Please allow us to clarify that our motivation stems from game theory and use adversarial policy to explore the empirical suboptimal behavior learned by MARL algorithms. If the victim’s policy reaches an equilibrium, there would be no vulnerability. However, real-world MARL algorithms often reach suboptimal equilibria, and the distance to the global optimum is hard to bound. Similarly, changes in environment dynamics and numbers of agents (as seen in transferable MARL methods [1-3]) can be unbounded, making existing work largely empirical.
>
> As requested, we have added a proof in response to Q5 to show that our local decomposition does not harm cooperation. However, such proofs often rely on strong assumptions, and the core contribution lies in our technical approach.
>
> We provide the proof in **Lines 331-357 and Appendix F** of the revised paper.
>
> > Q3: LLMs are used only to generate partner or opponent behaviors in certain evaluations, and there is no ablation study to assess their contribution to overall performance. (Weakness 5 & Question 3).
>
> Regarding the contribution of LLM-generated tactics, we conducted ablation studies in **Section 4.3**, showing that without adversarial tactic acquisition, our method performs significantly worse.
>
> We also evaluated the performance of various LLMs (GPT-4, Gemini 2.5 Pro, GPT-o3, GPT-3.5, and Gemini 1.5 Flash) in **Section 4.3 (Ablation of LLM-generated tactics)** and **Appendix E**. Our results show that once the LLM is sufficiently strong, the choice of model has minimal effect, with only minor scenario-specific differences, indicating that conventional LLMs have the common sense to generate effective tactics in cooperative–competitive games.
>
> We have clarified the ablation study and the impact of different LLMs in **Lines 498-502** and **Lines 513-516**.
>
> > Q4: The evaluated tasks (MPE, Overcooked, Atari) are relatively small and low-dimensional. Including experiments on more challenging MARL benchmarks (e.g., SMACv2) would better demonstrate scalability and robustness. (Weakness 4)
>
> We thank the reviewer for the suggestion regarding more challenging benchmarks. However, we **did not conduct experiments on MPE, Overcooked, or Atari**. Instead, we used **SMAC, SMACv2, and MAgent**, which involve high-dimensional state and action spaces, long-horizon coordination, and diverse task configurations, as detailed in **Section 4.1**.
>
> > Q5: Does the adaptive adversarial weighting mechanism maintain consistency between local and global objectives in cooperative settings? (Weakness 2 & Question 4)
>
> To ensure local-global consistency in our adversarial scene decomposition, we add all intermediate decisions (e.g., $b^i_t$ and $m^i_t$) to the input of the Q-function. The convergence of the Q-function thus ensures global optimality.
>
> We discuss the local-global consistency in **Lines 331-333** of the revised paper.
>
> ## References
>
> [1] Updet: Universal multi-agent reinforcement learning via policy decoupling with transformers, ICLR, 2021
>
> [2] Multi-agent policy transfer via task relationship modeling, 2022
>
> [3] Decompose a task into generalizable subtasks in multi-agent reinforcement learning, NIPS, 2023.

---

> ### Author Response · Authors · 2025-11-26
> **Invitation to reviewer-author discussion**
>
> Dear Reviewer,
>
> We are the authors of Submission 12609: "Adversarial Policy Transfer in Mixed Cooperative-Competitive Games". We’ve uploaded a point-to-point response to your comments, which we believe addresses all of your concerns.
>
> As the reviewer-author discussion period is going to end, we would greatly appreciate it if you could take a moment to share your thoughts on our rebuttal. Your feedback is valuable to us, and we’d be happy to further clarify any remaining points if needed.
>
> Best regards, Authors of Submission 12609

---

### Official Review · Reviewer_XeEr · 2025-11-04

**Soundness:** 3
**Presentation:** 2
**Contribution:** 3
**Rating:** 6
**Confidence:** 4

**Summary:**

The paper proposes a method combining LLM and Bayesian reasoning for adversarial policy transfer in multi agent reinforcement learning. Policies are modeled as probabilistic embeddings, with LLM generating dynamic strategy labels during training. Using a CTDE framework and Transformer backbone, the approach enables zero shot transfer to unseen tasks, aiming to enhance adaptability and generalization without fine tuning the LLM.

**Strengths:**

This paper provides a valuable exploration of combining large language models with Bayesian inference and effectively enhances agents’ ability to adapt strategies in complex environments. The proposed method is innovative, enabling the integration of natural language understanding into policy learning, which broadens the research perspective in this field. The overall structure of the paper is clear, and the technical details are supported by solid derivations and arguments. In addition, the experimental data and analysis are persuasive, thoroughly demonstrating both the strengths and limitations of the method.

**Weaknesses:**

The author uses LLMs to extract common paradigms from successful attack cases, but the paper does not provide detailed information about the sources or the sufficiency of these cases. Furthermore, the subsequent strategy iterations based on LLM tags rely entirely on the strategies given by the LLM in the first round, with only the weights of different tags being adjusted. Does this reliance on the initial strategy risk weakening the exploration of the strategy space and potentially lead to getting stuck in a local optimum?

**Questions:**

See weaknesses.

---

> ### Author Response · Authors · 2025-11-19
>
> We thank Reviewer XeEr for their appreciation of the novelty, technical rigor, clarity, and extensive experiments in our paper. Below is our detailed response to the reviewer’s concerns:
>
> ## Response
>
> > Q1: The author uses LLMs to extract common paradigms from successful attack cases, but the paper does not provide detailed information about the sources or the sufficiency of these cases.
>
> Regarding the sources, as stated in **Section 4.3 (Impact of LLM-generated tactics)** and **Appendix E**, we used GPT-4o to generate the adversarial tactics, alongside evaluations of other models, including Gemini 2.5 Pro, GPT-o3, GPT-3.5, and Gemini 1.5 Flash. Our experiments show that once the LLM reaches a certain level of strength (e.g., GPT-4o, Gemini 2.5 Pro, etc.), the choice of model has minimal impact, with only minor scenario-specific variations, indicating that these models share a common understanding of tactics in cooperative–competitive games.
>
> As for the sufficiency of LLMs, we also evaluated our attack's performance without using LLM-generated tactics. Our results show that the method significantly outperforms this ablated version, as detailed in **Section 4.3 (Ablation Study)**.
>
> Finally, we provide our experiment details, including exemplar prompts and outputs in **Appendix B**.
>
> In the revised version of the paper, we clarify the sources of LLMs in **Line 513-516** and the sufficiency of LLMs in **Line 498-502**.
>
> > Q2:  The subsequent strategy iterations based on LLM tags rely entirely on the strategies given by the LLM in the first round, with only the weights of different tags being adjusted.
>
> We would like to clarify that our method updates the adversarial tactics in each iteration, not just in the first round. In each iteration, the LLM provides a set of tactics and the weights of them for each training scenario, and the attack policy is trained from scratch with the new tactics. This ensures that exploration of the strategy space is not limited, and the method avoids getting stuck in local optima. This is discussed in **Lines 253-256** of the paper.
>
> In the revised version of the paper, we clarify the re-extraction and refinement of the LLM-generated adversarial tactics in **Line 265-268**.

---

> ### Author Response · Authors · 2025-11-26
> **Invitation to reviewer-author discussion**
>
> Dear Reviewer,
>
> We are the authors of Submission 12609: "Adversarial Policy Transfer in Mixed Cooperative-Competitive Games". We’ve uploaded a point-to-point response to your comments, which we believe addresses all of your concerns.
>
> As the reviewer-author discussion period is going to end, we would greatly appreciate it if you could take a moment to share your thoughts on our rebuttal. Your feedback is valuable to us, and we’d be happy to further clarify any remaining points if needed.
>
> Best regards, Authors of Submission 12609

---

### Meta-Review · Area_Chair_wDF8 · 2026-01-12

**Summary:**

Meta Review of Submission 12609
This paper introduces a framework for adversarial policy transfer in multi-agent reinforcement learning (MARL), utilizing Large Language Models (LLMs) to generate tactical labels that guide Bayesian strategy inference. The method aims to enable zero-shot transfer of adversarial policies to unseen cooperative-competitive environments through a bi-level optimization objective and policy distillation.

While the reviewers found the paper well-structured and the integration of LLMs with Bayesian reasoning innovative, several critical concerns led to a consensus for rejection. Reviewer XeEr noted that the strategy refinement process relies heavily on initial LLM tags, creating a risk of restricted exploration and convergence to local optima. Furthermore, Reviewer pdap highlighted a lack of theoretical rigor, specifically noting the absence of convergence guarantees for the bi-level optimization and the incremental nature of combining existing techniques like distillation and adversarial training. Reviewer ZaiY expressed concerns regarding the conceptual distinction between "adversarial exploitation" and general strategic superiority, suggesting the paper’s core definitions remain somewhat ambiguous.

Overall, despite the solid empirical performance on high-dimensional benchmarks like SMAC, the submission does not sufficiently address the fundamental risks of its limited search space or provide the theoretical stability required for the ICLR standard.

**Reviewer Concerns:**

Several weaknesses limit the paper’s contribution. Multiple reviewers highlight that the novelty is largely incremental, integrating existing components without a clear conceptual advance (pdap). The reliance on LLM-generated tactics raises concerns about transparency and rigor: the sources, sufficiency, and exact role of these tactics are not fully convincing, and their contribution is not cleanly isolated (XeEr, pdap). There is also notable confusion around the training vs. test-time optimization procedure and what makes the approach inherently adversarial rather than a general transferable strategy learner (ZaiY). Finally, the method lacks theoretical guarantees on convergence or stability, despite a complex bi-level adversarial setup (pdap).

**Reviewer Scores:**

It is hard to tell. The authors did address the reviewers' questions.

---

### Decision · Program_Chairs · 2026-01-26

Reject